# Cotransporter-mediated water transport underlying cerebrospinal fluid formation

Annette B. Steffensen[1], Eva K. Oernbo[1], Anca Stoica[1], Niklas J. Gerkau[2], Dagne Barbuskaite[3], Katerina Tritsaris[3], Christine R. Rose [2] & Nanna MacAulay [1]

Cerebrospinal fluid (CSF) production occurs at a rate of 500 ml per day in the adult human. Conventional osmotic forces do not suffice to support such production rate and the molecular mechanisms underlying this fluid production remain elusive. Using ex vivo choroid plexus live imaging and isotope flux in combination with in vivo CSF production determination in mice, we identify a key component in the CSF production machinery. The $Na^+/K^+/2Cl^-$ cotransporter (NKCC1) expressed in the luminal membrane of choroid plexus contributes approximately half of the CSF production, via its unusual outward transport direction and its unique ability to directly couple water transport to ion translocation. We thereby establish the concept of cotransport of water as a missing link in the search for molecular pathways sustaining CSF production and redefine the current model of this pivotal physiological process. Our results provide a rational pharmacological target for pathologies involving disturbed brain fluid dynamics.

[1] Department of Neuroscience, Faculty of Health and Medical Sciences, University of Copenhagen, Noerre Allé 14, 2200 Copenhagen, Denmark. [2] Institute of Neurobiology, Heinrich Heine University Duesseldorf, Universitaetsstrasse 1, 40225 Duesseldorf, Germany. [3] Department of Cellular and Molecular Medicine, Faculty of Health and Medical Sciences, University of Copenhagen, Noerre Allé 14, 2200 Copenhagen, Denmark. Correspondence and requests for materials should be addressed to N.M. (email: macaulay@sund.ku.dk)

The mammalian brain is bathed in the cerebrospinal fluid (CSF), which is continuously produced at a rate of approximately 500 ml fluid per day in the adult human[1]. Prior to exiting the brain, the CSF travels through the ventricular system and part of it re-enters the brain via the para-vascular route along the large arteries and penetrating arterioles[2,3]. The CSF is predominantly produced by the choroid plexus, an epithelial monolayer resting on highly vascularized connective tissue and located at the base of each of the four ventricles[4–7]. The molecular mechanisms underlying this choroidal fluid production remain unresolved. Dysregulation of CSF production or clearance may lead to brain water accumulation and raised intracranial pressure, as evident in patients with hydrocephalus. Hydrocephalus most commonly occurs as a consequence of obstructed CSF outflow, and is routinely treated by insertion of a ventriculo-peritoneal shunt diverting the excessive fluid from the ventricles into the peritoneal cavity in the abdomen[8]. However, in certain choroidal pathologies, such as choroid plexus hyperplasia, choroid plexus papilloma, and posthemorrhagic hydrocephalus, the increased intracranial pressure occurs, at least in part, from CSF overproduction[6,9,10]. The molecular mechanisms underlying the pathologic increase in CSF production remain elusive. Insight into the transport mechanisms underlying brain CSF accumulation could provide a rational therapeutic target to reduce this pathologic brain fluid accumulation.

The CSF production is generally assumed to take place by transport of osmotically active ions (e.g. sodium by the $Na^+/K^+$-ATPase[11,12]) followed by osmotically obliged, passive movement of water, partly via the water channel aquaporin 1 (AQP1) expressed at the luminal membrane of the choroid plexus[13,14]. However, several observations suggest that such a simple osmotic model may not be adequate: (1) The CSF production declined by a mere 20% in the AQP1 knock-out mice, partly ascribed to the 80% reduction of central venous blood pressure in these mice[15]. (2) With the known osmotic water permeability across the choroid plexus, detailed calculations have demonstrated that the osmolarity of the CSF must exceed that of the plasma by as much as 250 mOsm (in contrast to the measured difference in osmolarity of 5−10 mOsm[16,17]) in order for the CSF to be produced at the observed rate by simple osmosis[18]. (3) The choroid plexus has the ability to produce CSF against an oppositely directed osmotic gradient[18–21]. Taken together, conventional aquaporin-mediated osmotic water transport does not suffice to sustain the rates of CSF production consistently observed in mammals.

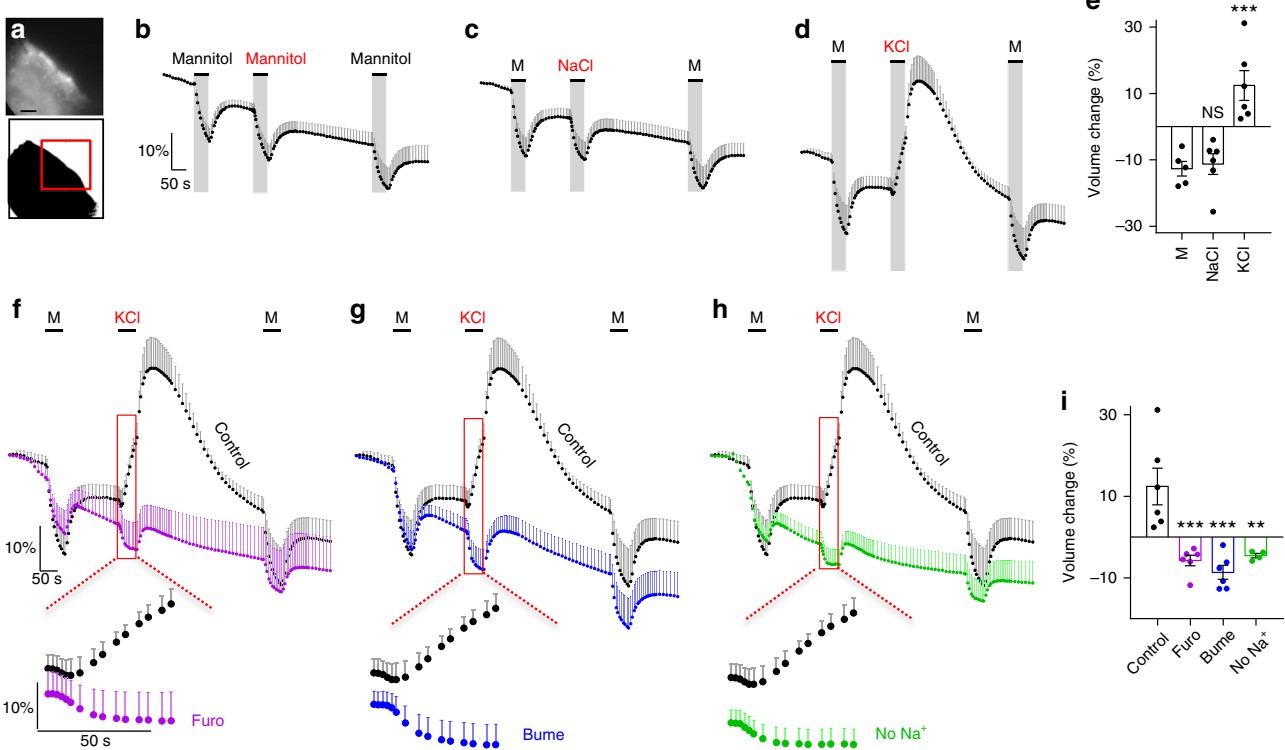

**Fig. 1** Cotransporter-mediated active water transport against an osmotic gradient. **a** Image of calcein fluorescence in mouse choroid plexus (CP, top) and the converted image (bottom) with indication of the region of interest (red box). Scale bar = 60 μm. **b** Volume changes of CPs challenged with a 100 mOsm gradient of mannitol (M), indicated by the extended bars (n = 5 CPs from five mice). **c** CPs challenged with a 100 mOsm gradient of first mannitol (M), then NaCl, and finally mannitol, indicated by the extended bars (n = 6 CPs from six mice). **d** CPs challenged with a 100 mOsm gradient of first mannitol (M), then KCl, and finally mannitol, indicated by the extended bars (n = 6 CPs from six mice). **e** Summary of the second volume change induced by identical osmotic gradients. **f** CPs challenged with 100 mOsm KCl alone (control, black symbols, data from **d**) or in the presence of 1 mM furosemide (purple symbols, n = 6 CPs from six mice). Inset magnifies the area of interest where furosemide blocks the KCl-induced swelling. **g** CPs challenged with 100 mOsm alone (control, black symbols, data from **d**) or in the presence of 10 μM bumetanide (blue symbols, n = 6 CPs from six mice). Inset magnifies the area of interest where bumetanide blocks the KCl-induced swelling. **h** CPs challenged with 100 mOsm KCl (control, black symbols, data from **d**) and in the absence of $Na^+$ in the test solution (green symbols, n = 4 CPs from four mice). Inset magnifies the area of interest where lack of $Na^+$ blocks the KCl-induced swelling. **i** Summary of the second volume change induced by KCl alone or with drug application/$Na^+$ omission. Error bars represent standard error of the mean and statistical significance was tested with one-way ANOVA followed by Tukey's multiple comparisons test. In panel **e**, the asterisks refer to a comparison to the mannitol-mediated shrinkage and in panel **i**, the asterisks refer to a comparison to the KCl-mediated swelling. **P < 0.01, ***P < 0.001, NS not significant

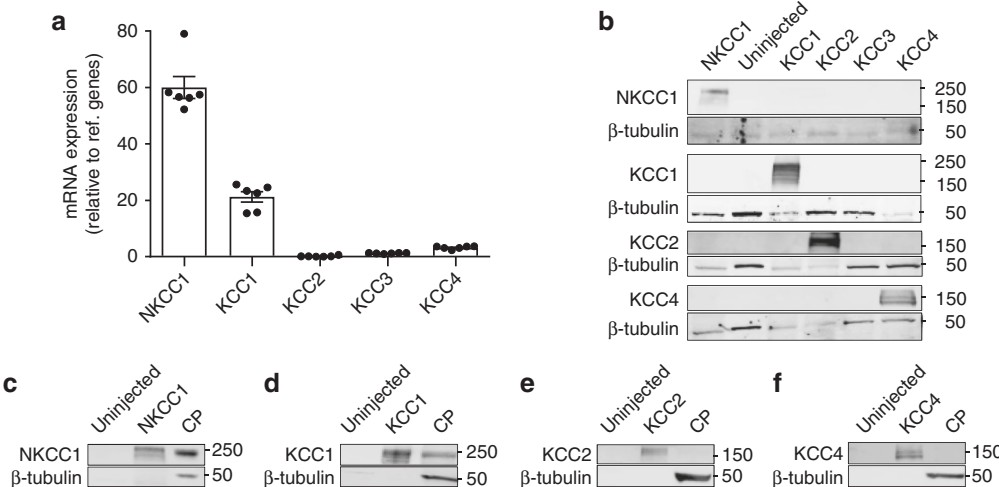

**Fig. 2** NKCC1 and KCC1 are expressed in mouse choroid plexus. **a** mRNA expression levels of NKCC1 and the four KCC isoforms in mouse choroid plexuses. The five target genes were normalized to two reference genes, GAPDH and H2AFZT, and presented as relative expression; NKCC1 = 60.0 ± 3.9, KCC1 = 21.2 ± 1.8, KCC2 = 0.2 ± 0.1, KCC3 = 1.2 ± 0.1, and KCC4 = 3.2 ± 0.2 (n = 6 mice), error bars represent standard error of the mean. **b** Representative western blots to verify antibody specificity. NKCC1 and KCC1-4 were expressed in *Xenopus laevis* oocytes and purified membranes from these and uninjected oocytes were exposed to SDS-PAGE followed by western blot (n = 3 experiments). **c–f** Western blots of lysates from mouse choroid plexus demonstrated expression of NKCC1 (**c**) and KCC1 (**d**), while KCC2 (**e**) and KCC4 (**f**) were not detected. Beta-tubulin was employed as loading control (n = 3 experiments)

A number of cotransporter proteins have the inherent ability to cotransport water along with the ions/solutes in the translocation mechanism (for review see refs. [18,22]). The coupling between water translocation and substrate transport takes place within the protein itself in a manner that permits water to be transported independently of, and even against, an osmotic gradient[23]. Examples of such water-translocating cotransporters are the Na$^+$/K$^+$/2Cl$^-$ cotransporter 1 (NKCC1) and the K$^+$/Cl$^-$ cotransporters (KCCs)[24–26]. Isoforms of these transport proteins have been detected in the choroid plexus epithelium[27–29], although their exact isoform distribution, relative expression, and membrane targeting remain largely unknown, as are their ability to transport water independently of an osmotic gradient in the choroid plexus tissue and their contribution to CSF production in vivo. In the present study, we introduce the water-translocating cotransporter, NKCC1, as the main contributor to CSF formation in the mouse choroid plexus.

## Results

**Choroidal cotransport of water against an osmotic gradient.** To determine if membrane transport mechanisms in the luminal membrane of choroid plexus carry an inherent ability to translocate water against an osmotic gradient, ex vivo mouse choroid plexus was monitored by live imaging during exposure to osmotic challenges. The acutely isolated choroid plexus was loaded with calcein-AM and the water movement determined as two-dimensional volume changes occurring as movement of the choroid plexus upon exposure to a hyperosmotic challenge of 100 mOsm (Fig. 1a). Three consecutive applications of 100 mOsm (100 mM) mannitol led to robust and reproducible shrinkage of choroid plexus (n = 5, Fig. 1b). This pattern was replicated with NaCl as the osmolyte (100 mOsm, 55 mM) during the second application (n = 6, Fig. 1c) while keeping mannitol application as a reference for the first and last osmotic challenge. In contrast, application of a 100 mOsm hyperosmolar challenge introduced via addition of KCl (55 mM) generated an abrupt choroid plexus volume increase (n = 6, Fig. 1d), indicating that water was transported *into* the choroid plexus despite the large oppositely directed osmotic gradient. It should be noted that the volume

increase occurred instantaneously prior to significant changes in intracellular parameters[22,25]. Summarized data from the second osmolyte application illustrate that the volume decrease obtained upon application of 100 mOsm mannitol (12.7 ± 2.2%, n = 5) or NaCl (11.3 ± 3.1%, n = 6) P = 0.96, df = 14, q = 0.4) reached comparable levels (Fig. 1e), while application of an identical osmotic challenge given in the form of KCl produced a volume *increase* of 12.4 ± 4.5%, n = 6 (one-way ANOVA followed by Tukey's multiple comparisons test, P = 0.0006, df = 14, q = 6.9, Fig. 1e). These results indicate that choroid plexus contains transport mechanisms capable of transporting water independently of, and even against, the direction of an applied osmotic gradient. To determine the involvement of cation-chloride cotransporters (CCCs) in the K$^+$-mediated transport of water against an experimentally applied osmotic gradient, we continued the experimental series described above with the inclusion of pharmacological agents. Application of furosemide (1 mM inhibits both KCCs and NKCCs[30]) completely blocked the K$^+$-mediated choroid plexus volume increase: The tissue responded in a passive manner, as when challenged with mannitol as the osmolyte (shrinkage of 5.7 ± 1.3%, one-way ANOVA followed by Tukey's multiple comparisons test, P = 0.0006, df = 18, q = 6.9, n = 6, Fig. 1f). For illustrative purposes, the volume traces obtained in control solution in the previous figure (Fig. 1d) are included in black. Inclusion of bumetanide (10 μM inhibits NKCC1[31], 8.6 ± 1.7% shrinkage, one-way ANOVA followed by Tukey's multiple comparisons test, P = 0.0001, df = 18, q = 8.0, n = 6, Fig. 1g) or removal of Na$^+$ from the test solution (equiosmolar replacement with choline; 4.5 ± 0.5% shrinkage, one-way ANOVA followed by Tukey's multiple comparisons test, P = 0.0037, df = 18, q = 5.8, n = 4, Fig. 1h) likewise abolished the K$^+$-mediated water accumulation. Data are summarized in Fig. 1i. Increased [K$^+$]$_e$ thus promoted inwardly directed ion transport by NKCC1, which by its ability to cotransport water during its translocation mechanism contributed to intracellular water accumulation against a substantial osmotic gradient. Cotransporter-mediated water transport is thereby indeed able to move water across the choroid plexus membrane in a manner independent of an osmotic gradient.

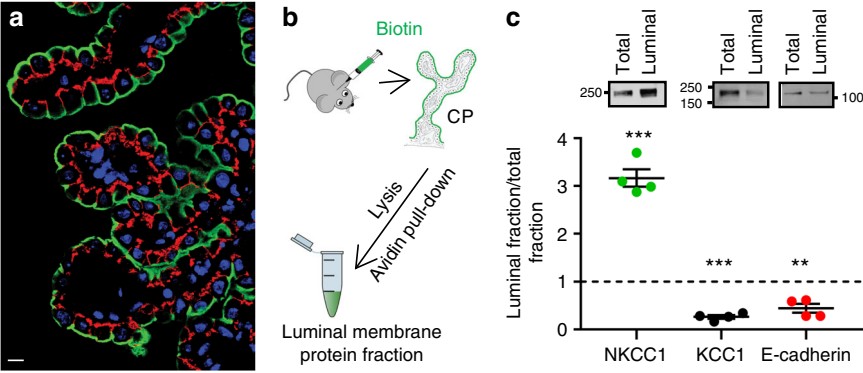

**Fig. 3** NKCC1 is located at the luminal membrane facing the ventricles. **a** Immunostaining of mouse choroid plexus illustrated expression of NKCC1 in the membrane facing the lumen (green) with E-cadherin (red) as a basolateral marker and nuclei in blue. Scale bar = 10 μm. Previously unpublished image generously provided by Dr. Jeppe Praetorius. **b** Schematic illustration of the luminal surface biotinylation of mouse choroid plexus in situ. **c** Representative western blots showing expression of NKCC1, KCC1, and E-cadherin in total tissue fraction ($F_t$) and purified biotinylated (luminal) membrane protein fraction ($F_l$). Below is depicted the quantification of protein abundance represented as luminal membrane/total protein fraction ($n = 4$ mice). NKCC1 was expressed significantly higher in the purified luminal membrane fraction while KCC1 and E-cadherin were predominantly in the total fraction. Error bars represent standard error of the mean and statistical significance was determined with one-sample $t$ test and comparison to equal distribution; $F_l/F_t = 1$ and the asterisks refer to a value significantly different from 1. **$P < 0.01$, ***$P \leq 0.001$

**NKCC1 and KCC1 are expressed in choroid plexus.** To determine the isoform-specific expression profile of NKCC1 and the KCCs in mouse choroid plexus, we initially performed quantitative PCR on mRNA purified from acutely isolated tissue. Normalization of the obtained quantities of mRNA encoding NKCC1 and KCC1-4 to two reference genes illustrated a robust expression of NKCC1 and, although to a lesser extent, KCC1, while KCC2-4 mRNA expression was minor or below detection limit ($n = 6$, Fig. 2a). Prior to determination of the protein expression of the CCCs in choroid plexus tissue, we verified that the employed antibodies exclusively recognized their targeted isoform. To this end, NKCC1 and KCC1-4 were individually expressed in *Xenopus laevis* oocytes and purified membranes containing each isoform were exposed to SDS-PAGE followed by western blot with antibodies targeting each of these transport proteins. As evident from Fig. 2b, the employed antibodies recognized only their respective target protein with no isoform cross-reaction ($n = 3$). We were, however, unsuccessful with all five KCC3 antibodies tested, none of which properly recognized KCC3, either due to poor epitope recognition or lack of KCC3 expression in the oocytes. Western blotting of lysates from mouse choroid plexus demonstrated robust immunoreaction with antibodies targeting NKCC1 and KCC1, while KCC2 and KCC4 were below detection limit in this tissue ($n = 3$, Fig. 2c, f). None of the five tested KCC3 antibodies provided indications of immunoreactivity in the choroid plexus tissue, indicating either lack of choroidal expression of this isoform or poor epitope recognition. Taken together, mRNA and protein analysis illustrate robust expression of NKCC1 and KCC1 in the mouse choroid plexus and negligible expression of KCC2-4.

**NKCC1 is localized to the luminal membrane in choroid plexus.** Localization of NKCC1 and KCC1 to the luminal membrane of the choroid plexus is required in order for these to directly transport water into the ventricular lumen. Immunohistochemical staining of mouse choroid plexus verified the localization of NKCC1 at the luminal membrane (green, Fig. 3a). Three antibodies directed towards KCC1 produced only diffusive, and therefore inconclusive, staining in choroid plexus tissue. To determine the membrane targeting of KCC1 in an alternative fashion, we performed surface biotinylation of the choroid plexus.

Rupture of the choroidal tissue during the isolation procedure yielded undesirable access of the biotin to the basolateral membrane. Instead, selective biotinylation of the lumen-facing choroidal membrane was obtained by ventricular delivery of biotin in the intact animal, prior to isolation of the choroid plexus and purification of the biotinylated proteins (see Fig. 3b for a diagram of the procedure). Densitometric analysis of the western blots of the biotinylated membrane protein fraction relative to the total membrane protein fraction illustrated NKCC1 enrichment in the biotinylated (luminal) membrane fraction ($F_l$) compared to the total membrane fraction ($F_t$) (Fig. 3c, left panels, $F_l/F_t = 3.2 \pm 0.2$, $n = 4$, one-sample $t$ test and comparison to equal distribution; $F_l/F_t = 1$, $P = 0.001$, $t = 11.82$, df = 3). In contrast, abundance of KCC1 was significantly lower in the biotinylated fraction (Fig. 3c, middle panels, $F_l/F_t = 0.3 \pm 0.0$, $n = 4$, one-sample $t$ test and comparison to equal distribution; $F_l/F_t = 1$, $P < 0.001$, $t = 20.1$, df = 3) and resembled the pattern obtained with E-cadherin, a basolaterally located membrane protein[32] (Fig. 3c, right panels, $F_l/F_t = 0.4 \pm 0.1$, $n = 4$, one-sample $t$ test and comparison to equal distribution; $F_l/F_t = 1$, $P = 0.009$, $t = 6$, df = 3). Therefore, NKCC1, not KCC1, is localized to the luminal membrane of choroid plexus, from which it could indeed participate in CSF production.

**NKCC1 is poised for outwardly directed transport.** The NKCC1 transport is inwardly directed in most cell types[33] while production of CSF necessitates outwardly directed transport of ions and water. To reveal the choroidal NKCC1 transport direction, we determined the ion concentrations of $Na^+$, $K^+$, and $Cl^-$ in choroid plexus epithelial cells and CSF of mice. CSF was extracted from anesthetized and artificially ventilated mice by a glass capillary inserted in cisterna magna. Immediately thereafter, each mouse was sacrificed, choroid plexus isolated ($n = 4$), and the ion concentrations determined with flame photometry ($Na^+$ and $K^+$) or by colorimetry ($Cl^-$). The obtained ion concentrations (CSF: $150 \pm 1$ mM $Na^+$, $3 \pm 0$ mM $K^+$, $100 \pm 6$ mM $Cl^-$ and in the choroid plexus epithelial cells: $31 \pm 5$ mM $Na^+$, $141 \pm 12$ mM $K^+$, and $35 \pm 9$ mM $Cl^-$, $n = 4$) illustrate that the intracellular $Na^+$ and $Cl^-$ concentrations are substantially higher than the 5–15 mM range usually observed in most mammalian cells[34]. Calculation of the Gibbs free energy using the obtained ion

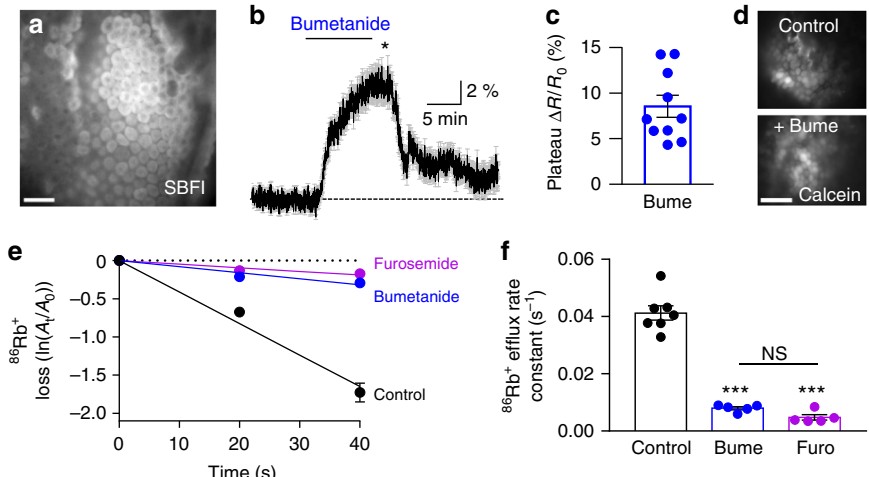

**Fig. 4** NKCC1 is poised for outward transport in the luminal membrane. **a** Image of SBFI fluorescence of the choroid plexus taken with wide-field fluorescence microscope. Scale bar = 40 μm. **b** Representative sodium signals of choroid plexus during 10 min baseline, 10 min application of the NKCC1 inhibitor bumetanide (10 μM), and 20 min washout. Gray lines represent the SBFI ratio obtained from 40 single cells during one experiment, black line represents an average of these cells. **c** Average bumetanide-induced increase in SBFI ratio of choroid plexus cells, indicative of increased intracellular sodium concentration ($n = 10$ CPs from ten mice). **d** Representative images of calcein fluorescence of the choroid plexus either after 10 min in aCSF (top panel) or after 10 min of bumetanide treatment (bottom panel, time point marked as * in **b**). Scale bar = 40 μm. **e** Loss of $^{86}Rb^+$ from choroid plexus as a function of time in control settings (black, $n = 7$ CPs from seven mice) or with treatment of either bumetanide (20 μM, blue, $n = 5$ CPs from five mice) or furosemide (1 mM, violet, $n = 5$ CPs from five mice). Y-axis is the natural logarithm of amount of $^{86}Rb^+$ left in choroid plexus at time $t$ ($A_t$) divided by the amount at time 0 ($A_0$). **f** Efflux rate constants for $^{86}Rb^+$ in control ($0.041 \pm 0.003 \, s^{-1}$, $n = 7$ CPs from seven mice), in the presence of bumetanide ($0.008 \pm 0.001 \, s^{-1}$, $n = 5$ CPs from five mice), or furosemide ($0.005 \pm 0.001 \, s^{-1}$, $n = 5$ CPs from five mice). Error bars represent standard error of the mean and statistical significance was tested with one-way ANOVA followed by Tukey's multiple comparisons test and the asterisks above the bars indicate comparison to the control while the comparison between the two test solutions is indicated with a line above the relevant bars. ***$P < 0.0001$, NS not significant ($P = 0.526$)

concentrations (see Methods) yields $\Delta G = 448 \, J \, mol^{-1}$, which indicates an outwardly directed transport of NKCC1 in the lumen-facing membrane of mouse choroid plexus epithelial cells under physiological conditions. To experimentally determine the transport direction of NKCC1 in choroid plexus epithelial cells, we performed wide-field $Na^+$ imaging on ex vivo choroid plexus, loaded with SBFI-AM (sodium-binding benzofuran isophthalate acetoxymethyl ester) (Fig. 4a). In choroid plexus kept in control aCSF, $[Na^+]_i$ was stable and did not undergo detectable fluctuations (Fig. 4b). Upon inhibition of NKCC1 (10 μM bumetanide), however, the SBFI fluorescence ratio increased by $8.5 \pm 1.2\%$ within 10 min ($n = 10$, Fig. 4b, c), revealing an increase in $[Na^+]_i$. This increase indicates outwardly directed NKCC1-mediated $Na^+$ transport. Upon 20 min washout of bumetanide, the $[Na^+]_i$ returned towards baseline ($2.4 \pm 0.6\%$, $n = 5$, Fig. 4b), reflecting re-establishment of $Na^+$ export by the NKCC1. Bumetanide did not compromise cell viability (confirmed by uptake of calcein-AM as a marker of cell health[35], Fig. 4d, $n = 4$ of each condition), indicating that the observed increase in $[Na^+]_i$ did not result from unspecific influx of $Na^+$, but was indeed due to blocking of NKCC1 transport activity. The unique transport direction of NKCC1 was verified with efflux experiments with $^{86}Rb^+$, a congener for $K^+$, from pre-loaded ex vivo choroid plexus. The $^{86}Rb^+$ efflux decreased approximately 85% after exposure to either bumetanide (20 μM, $n = 5$, one-way ANOVA followed by Tukey's multiple comparisons, $P < 0.0001$, df = 14, $q = 17.6$) or furosemide (1 mM, $n = 5$, one-way ANOVA followed by Tukey's multiple comparisons, $P < 0.0001$, df = 14, $q = 19.3$) (Fig. 4e, f), indicating that the $K^+$ efflux was predominantly orchestrated by NKCC1. Together, these results show that the ion concentrations of choroid plexus epithelial cells uniquely dictate outwardly directed transport of NKCC1, which thus could act as a contributor to CSF production.

**NKCC1 significantly contributes to CSF production in vivo**. To determine the contribution of NKCC1 to CSF production in vivo, we performed experiments on anesthetized mice placed in a stereotaxic frame during ventriculo-cisternal perfusion (modified from refs. [15,36]). In this experimental approach, aCSF containing fluorescent dye (dextran) is perfused via a cannula through the lateral ventricle of the mouse ($0.7 \, \mu l \, min^{-1}$) with simultaneous fluid collection by a glass capillary from the cisterna magna at 5-min intervals (Fig. 5a). The dilution of the fluorescent dye represents the rate of CSF production. As our experimental protocol was based on employing each mouse as its own control, it was an absolute requirement to record a sustained rate of CSF production throughout the prolonged experimental procedure (125 min). To obtain such standard, the animals were artificially ventilated, their heart rate, respiratory partial pressure of carbon dioxide, and arterial oxygen saturation were monitored, and their core temperature maintained (see Methods). A representative time control experiment is depicted in Fig. 5b, in which the dextran gradually appeared in the CSF samples and a stable dilution obtained after approximately 40 min. The last two samples prior to the solution change (60 min, marked in red in Fig. 5b) were employed to calculate the CSF production rate of $0.66 \pm 0.02 \, \mu l \, min^{-1}$ ($n = 18$, Fig. 5b inset). A similar CSF production rate ($0.60 \pm 0.02 \, \mu l \, min^{-1}$, $n = 6$, $P = 0.21$, $t = 1.3$, df = 22, $t$ test) was obtained in a set of animals anesthetized with isoflurane rather than ketamine/xylazine (see Methods). The vehicle (DMSO)-containing aCSF delivered to the lateral ventricle was replaced with new DMSO-containing aCSF at the time point marked "solution change". After 60 min additional sample time, the last two sample points (blue in Fig. 5b) were normalized to the original baseline (the red points) revealing a slight (although non-significant) drop of baseline to $92.2 \pm 3.3\%$ during the course of the experiment ($n = 6$, one-way ANOVA followed by Tukey's

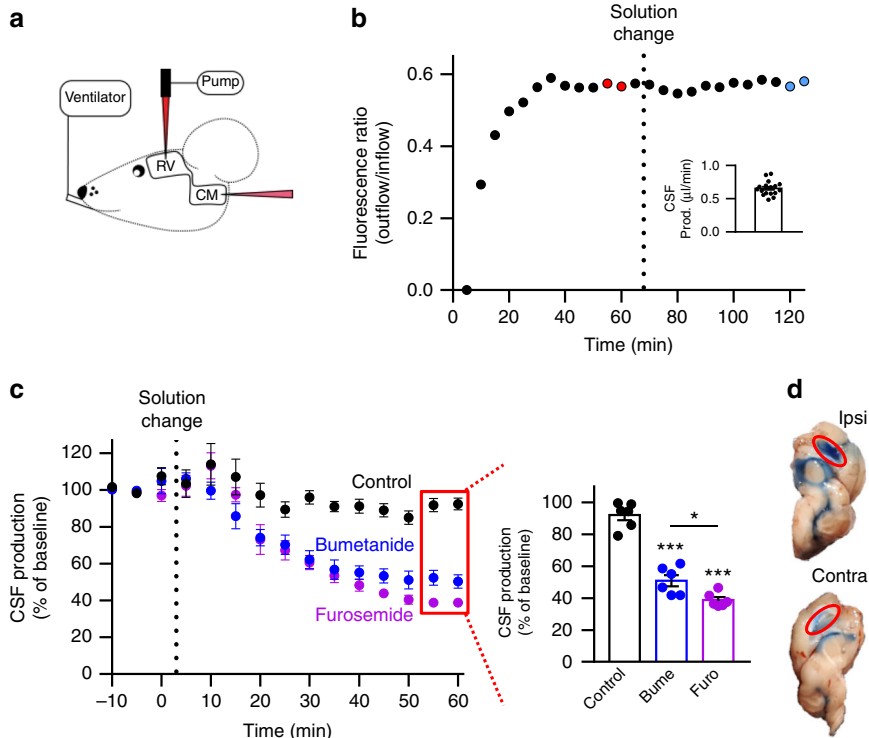

**Fig. 5** NKCC1 acts as a significant contributor to in vivo CSF production. **a** Schematic drawing of the infusion of aCSF containing dextran (dark red cannula) into the right lateral ventricle (RV) and collection of the diluted dextran (light red cannula) at cisterna magna (CM) of a ventilated mouse. **b** Representative time course of the fluorescence ratio of dextran (outflow/inflow) during a control ventriculo-cisternal perfusion. After stable baseline (60 min), the dextran/DMSO aCSF-solution was changed within 5–10 min. Inset depicts the average CSF production rate ($n = 18$ mice). **c** Summarized data from ventriculo-cisternal perfusion illustrating CSF production (percentage of baseline) as a function of time. Data normalized to the average of the two last samples before solution change. Control perfusion with the DMSO-vehicle is shown in black ($n = 6$ mice), treatment with bumetanide in blue ($n = 6$ mice), and furosemide (n = 6 mice) in purple. Inset illustrates the summarized CSF production rates after 60 min exposure to vehicle (black), bumetanide (blue), or furosemide (purple) normalized to own control. **d** Mid-sagittal section of a mouse brain after unilateral injection of Evans Blue revealed staining mainly in the injected, right lateral ventricle (top panel) compared to the contralateral ventricle (bottom image). Error bars represent standard error of the mean and statistical significance was tested with one-way ANOVA followed by Tukey's multiple comparisons test with asterisks above the bars indicating comparison to control perfusion and comparison between test solutions indicated by lines above the respective bars. *$P < 0.05$, ***$P < 0.001$

multiple comparisons, $P = 0.06$, $t = 2.4$, df = 5, Fig. 5c), demonstrating rather stable CSF production throughout the experimental procedure. To determine the quantitative contribution of NKCC1 to CSF production, aCSF including bumetanide (100 µM) was introduced with the solution change, which reduced the CSF production to $51.0 \pm 3.5\%$ of baseline ($n = 6$, one-way ANOVA followed by Tukey's multiple comparisons, $P < 0.0001$, df = 15, $q = 13.9$, Fig. 5c). Inclusion of the NKCC1/KCC inhibitor furosemide (2 mM) reduced the CSF production to $38.8 \pm 1.8\%$ of baseline ($n = 6$, one-way ANOVA followed by Tukey's multiple comparisons, $P < 0.0001$, df = 15, $q = 18.0$, Fig. 5c), slightly more than that observed with bumetanide (one-way ANOVA followed by Tukey's multiple comparisons, $P = 0.03$, df = 15, $q = 4.1$). Based on these results, it is evident that NKCC1 is a substantial contributor to the molecular machinery underlying CSF production in an in vivo experimental setting. While this dilution method reflects CSF production of all origins (all four choroid plexuses in addition to trans-capillary fluid production), delivery of a pharmacologic inhibitor via the cannula placed in one lateral ventricle may not reach the transport mechanisms expressed in the choroid plexus at the base of the contralateral ventricle. To obtain an estimate of the reach of such one-sided drug delivery, Evans blue was infused into one lateral ventricle in a manner mimicking the experimental approach above, prior to isolation of the brain. As evident in Fig. 5d, staining was predominantly observed in the perfused lateral ventricle with little staining in the contralateral one. We therefore predict partial inhibition of the cotransporters in the contralateral choroid plexus and that our data thus represent an underestimate of the role of NKCC1 in CSF production.

**Live imaging shows NKCC1-mediated CSF production**. To obtain a swift and less-invasive manner of revealing NKCC1-mediated CSF production, we developed an in vivo imaging strategy based on the LI-COR Pearl Trilogy small animal imaging system. This method allows imaging of anesthetized animals immediately after lateral ventricular delivery of a fluorescent dye (IRDye 800CW carboxylate). The ventricular dye redistribution is employed as a proxy of CSF production, although diffusion of the dye is expected to contribute to its redistribution. Figure 6a illustrates the head of a white mouse (obtained as a white light image) immediately after ventricular delivery of the fluorescent probe (superimposing of pseudo-color fluorescence). The red square indicates the area of interest, in which the dye intensity is determined as a function of time (representing movement of CSF) (Fig. 6b, d). After 5-min pre-injection of inhibitor (or vehicle) the fluorescent dye was injected into one lateral ventricle along with inhibitor (or vehicle)-injection, and the mouse rapidly placed in the LI-COR Pearl (exactly 1 min lapse from injection to first image acquisition); see Fig. 6d for representative images. The

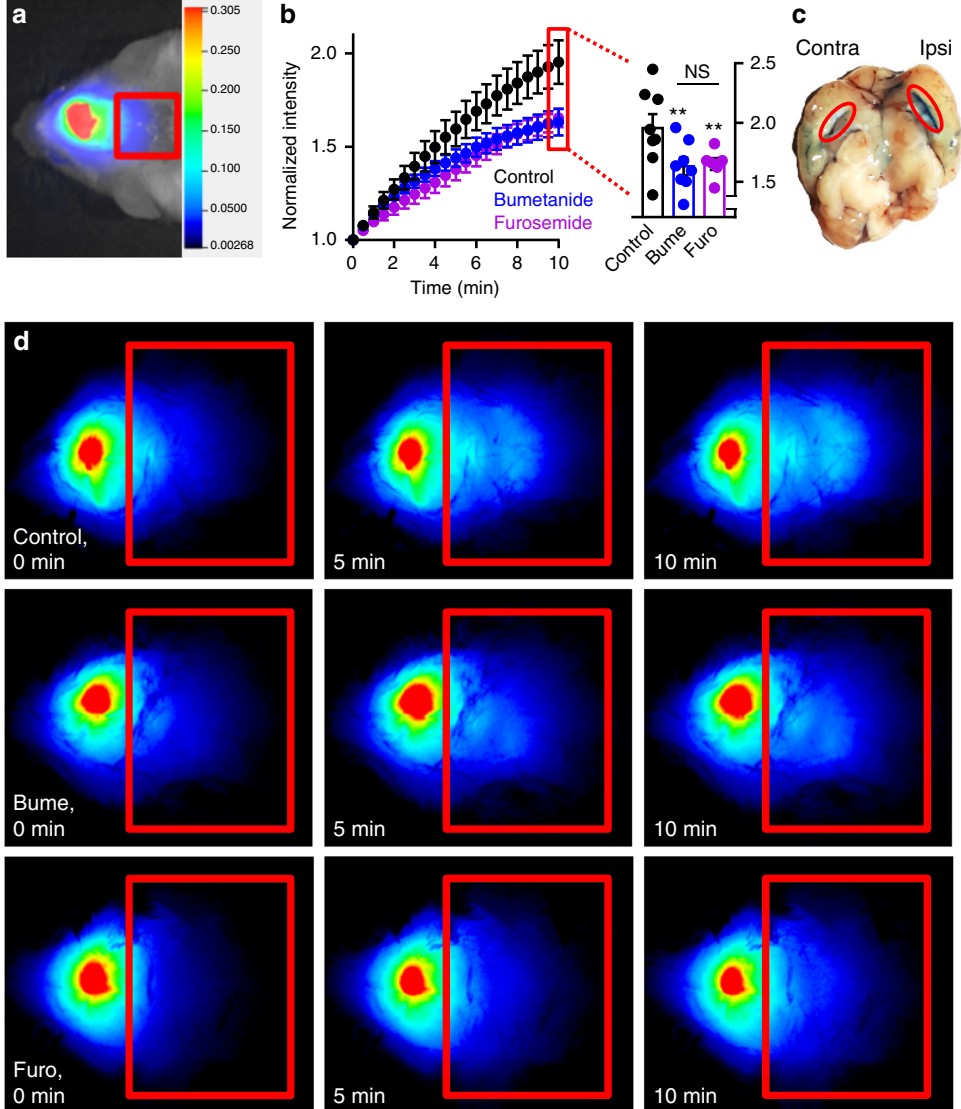

**Fig. 6** Live imaging determines significant NKCC1 contribution to in vivo CSF production. **a** Illustration of pseudo-color fluorescence superimposed on a white light image from a white mouse after ventricular injection of 10 μM IRDye 800CW carboxylate dye. The dye content is quantified in the red box, placed in line with lambda. The intensity scale is arbitrary units and applies to all images. **b** Intensity of images were quantified and normalized to the first image (0 min) for control (vehicle, black), bumetanide (blue), or furosemide (purple) and plotted as a function of time. Inset illustrates summarized data at the 10-min time point; the fluorescence intensity of control mice reached 1.95 ± 0.12 ($n = 8$ mice), furosemide-treated 1.65 ± 0.05 ($n = 6$ mice), and bumetanide 1.63 ± 0.07 ($n = 8$ mice). **c** Unilateral injection of Evans Blue revealed staining mainly in the injected, right lateral ventricle (red oval) compared to the contralateral ventricle. **d** Representative images at 0, 5, and 10 min for control condition (top panels), bumetanide treatment (middle panels), and furosemide treatment (bottom panels). Error bars represent standard error of the mean and statistical significance was evaluated with two-way ANOVA (RM) followed by Tukey's multiple comparisons test. Asterisks above the bars indicate comparison to control while comparisons between test solutions are indicated with a line above the respective bars. **P < 0.01, NS: not significant ($P = 0.98$)

fluorescence intensity recorded in the area of interest was normalized to the intensity of the first image, summarized, and illustrated as a function of time (Fig. 6b). Inclusion of bumetanide (100 μM, $n = 8$, two-way analysis of variance (ANOVA) (repeated measures (RM)) followed by Tukey's multiple comparisons test, $P = 0.0018$, df = 399, $q = 4.9$) or furosemide (2 mM, $n = 6$, two-way ANOVA (RM) followed by Tukey's multiple comparisons test, $P = 0.0083$, df = 399, $q = 4.2$) significantly decreased the movement of dye compared to control, indicative of NKCC1-mediated movement of ventricular fluorescence and thus CSF flow. Of note, injection of Evans blue, in a manner and quantity mimicking the experimental approach above, predominantly stained the ipsilateral ventricle (Fig. 6c), indicative of a putative underestimation of the inhibitor-sensitive movement of the fluorescent dye.

## Discussion

Here, we have demonstrated, by complementary ex vivo and in vivo experimentation, that the high production rate of CSF is sustained by NKCC1 via its inherent ability to cotransport water along with its directional ion translocation in a manner independent of osmotic driving forces. This unconventional means of fluid secretion underlying CSF production represents a paradigm shift in the field and provides a long-needed rational therapeutic target towards brain pathologies involving disturbances in brain water homeostasis and increased intracranial pressure.

The limitations of a conventional osmotic model for CSF production are apparent from three independent lines of evidence: The minimal effects of genetic deletion of AQP1, the low osmotic water permeability of the epithelium, and the ability of

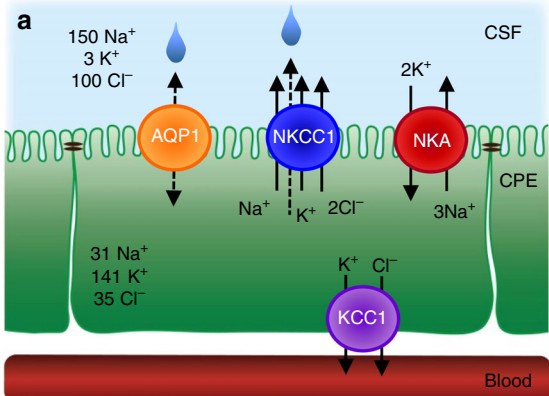
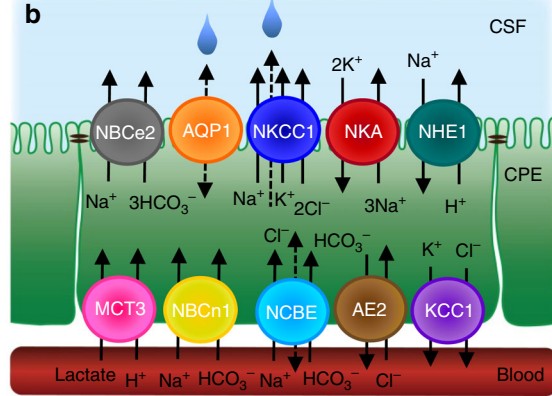

**Fig. 7** Schematic drawing of transporters in a choroid plexus epithelial cell. **a** The drawing depicts the selective expression of NKCC1 at the luminal membrane and indicates its unique outward transport direction, due to the high intracellular concentration of $Na^+$ and $Cl^-$ in this tissue (ion concentrations given in mM). The NKCC1-mediated cotransport of water is indicated with a dashed arrow. The $Na^+/K^+$-ATPase and AQP1[14] are indicated on the luminal membrane of the choroid plexus epithelium (CPE), while KCC1 is localized to the basolateral membrane facing the vascular compartment (this study). **b** The drawing includes the many other coupled transporters localized to the choroidal epithelial membranes[1]. Note that NCBE may also be referred to as NBCn2

the choroid plexus epithelium to transport water uphill against a transepithelial osmotic gradient. Firstly, discovery of AQP1 in the lumen-facing membrane of choroid plexus initially promoted its implication in CSF production[13,15,27]. A slightly reduced CSF production was observed in AQP1$^{-/-}$ mice, although accompanied by a severe drop in venous blood pressure in these animals[15]. With the general view that CSF originates from the vascular compartment[37,38], taken together with a fivefold higher arterial flow in choroid plexus compared to that of the brain as a whole[12], such AQP1-dependent reduction in blood pressure[15] is anticipated to contribute to the reduced CSF production observed in AQP1$^{-/-}$ mice. Secondly, it has been assumed that CSF production takes place via conventional osmotic water transport following a ventricular build-up of osmotic particles through activity of choroidal membrane transport mechanisms, such as the $Na^+/K^+$-ATPase, cotransporters, and/or ion channels[1,4,13]. Concerted action of these transport mechanisms renders the CSF approximately 5−10 mOsm hyperosmolar relative to plasma[16,17]. With the established water permeability of the choroid plexus epithelium[39], calculations dictate that the transepithelial osmotic gradient entailed to sustain the high CSF production rate would need to be about 50 times greater than that observed[18]. Thirdly, while the rate of CSF production indeed increases with raised ventricular osmolarity, CSF production continues even when the ventricular fluid is hyposmotic to plasma[18–21]. In goat, the in vivo CSF production continued despite opposing osmotic gradients of up to 100 mOsm[20].

Simple osmotically obliged water entry into the ventricle is therefore insufficient to sustain the well-established rate of CSF production and here we propose cotransport of water as the molecular mechanism underlying CSF production. Our research group earlier demonstrated that cotransporters function as unconventional water transporters carrying 300–500 water molecules along with the transported solutes per transport cycle[18,22], rendering the combined solute and water transport near-isotonic. NKCC1 is prominently featured among these and has in complementary cell systems been demonstrated to cotransport water[24,26]. Here, we reveal that the mouse choroid plexus displays the ability to transport water against a significant osmotic gradient in a $K^+$-induced, NKCC1-mediated manner. While the high $[K^+]_o$ strongly favors inward transport by NKCC1, and thus robust cell swelling, the high $[Na^+]_o$ is,

although to a lesser degree, likely to do so as well. However, with the low apparent affinity of NKCC1 for $K^+$[40], the inwardly directed transport under conditions of high $[Na^+]_o$ and low $[K^+]_o$ and the resulting NKCC1-mediated cell swelling will be limited and thus likely masked by the parallel osmotically induced cell shrinkage. Our data align with an earlier study on salamander choroid plexus, performed with ion-sensitive microelectrodes, which, however, relied on the KCC for movement of water independently of the prevalent osmotic gradient[25]. It should be emphasized that the NKCC1-mediated component of CSF production is a molecular property within the transport protein and not a result of unstirred layers around the protein or changes in intracellular osmolarity[22].

Our expression studies of mRNA and protein were conducted on pooled tissue obtained from the lateral and the fourth choroid plexuses from mouse (to obtain sufficient tissue) and revealed robust expression of NKCC1. The observation of NKCC1 localization in the lumen-facing membrane of mouse aligns with reports on human[27] and rat[41] choroid plexus. We observed KCC1 expression at the mRNA[42,43] and protein level and localized it to the basolateral membrane, while KCC2-4 mRNA and protein expression was negligible or below detection limit in our samples. Lack of KCC2 mRNA and protein in choroid plexus confirms previous studies[42,44], whereas a few reports have assigned KCC3 and KCC4 protein to choroidal tissue[28,29] despite absence or downregulation after birth of mRNA encoding these isoforms (this study and ref. [43]). We speculate that the discrepancy might arise from insufficient antibody specificity, by the developmental changes in KCC mRNA expression[43], and/or the distinct expression profile of the different choroid plexuses[45]. We are, nevertheless, left with the conclusion that in adult mouse choroid plexus, NKCC1 is expressed in the membrane facing the ventricular lumen and KCC1 in the basolateral membrane. The remarkably high intracellular concentrations of $Na^+$ (~30 mM) and $Cl^-$ (~35 mM) in the mouse choroidal epithelium align with values obtained in rat[1,46] and suffice to promote outwardly directed NKCC1 transport. This atypical NKCC1 transport direction was experimentally verified by complementary techniques, aligns with rat tissue[47], and supports the notion that NKCC1 is uniquely well-suited to participate in CSF production via its outwardly directed transport in the choroid plexus epithelium (see summary Fig. 7a). This is in contrast to other epithelia and cell types, in which NKCC1 is generally poised towards

inwardly directed transport[48,49]. NKCC1 is therefore detected on the basolateral membrane of other secretory epithelia[33].

Optimal CSF production relies on sustained physiological parameters in the anesthetized animal. We therefore propose that to reliably measure the rate of CSF production and assign quantitative contributions of different transport mechanisms, it is crucial to make every effort to maintain core body temperature and ensure proper artificial ventilation of the animal, and hence physiological values for arterial partial pressure of carbon dioxide and oxygen, arterial pH, heart rate, and blood pressure. Of note, ventilated mice had a significantly higher CSF production rate of $0.66 \pm 0.02\,\mu l\,min^{-1}$ than determined earlier in non-ventilated mice $(0.38 \pm 0.02\,\mu l\,min^{-1}$ [15]) and in our unpublished pilot experiments. Determination of CSF production by the ventriculo-cisternal perfusion method relies on dilution of ventricularly delivered dextran by the newly produced CSF. This method, however, cannot distinguish between CSF formed by the choroid plexus (presumably the majority[4–7]) and that entering the ventricles via the ependymal cell layer following its secretion across the capillary wall in the reminder of the brain[50]. The endothelial expression of NKCC1 is negligible[41], and the bumetanide-sensitive fraction of the CSF production is therefore assigned to the highly NKCC1-expressing choroid plexus. Inhibition of NKCC1 (upon intraventricular delivery of inhibitor) decreased CSF production by 50% in mice, the quantitative importance underscored by a similar finding in dogs[51]. Notably, intravenous delivery of CCC inhibitors consistently fail to affect CSF production rate[9,52,53], in alignment with the luminal membrane expression of NKCC1 illustrated in summary Fig. 7a. The ventricular perfusion rate was set to approximately the same speed as the CSF production rate, thereby expecting a 1:1 dilution of the applied inhibitors. While we therefore increased the inhibitor concentration in the in vivo experimentation compared to that employed ex vivo, we cannot exclude a lower concentration of inhibitor reaching the choroid plexus epithelium at the base of the ventricles, due to the continuous production of CSF and the ensuing wash out. Taken together with limited reach of the applied inhibitor to the contralateral ventricle (illustrated with Evans blue), we predict that our results may underestimate the quantitative contribution of NKCC1 to CSF production and that NKCC1 may represent even more than the observed half of the CSF production machinery.

While in larger animal models, it would be technically feasible to place cannulas in both lateral ventricles and apply double lateral perfusion (to determine the full contribution of NKCC1), it is highly likely that other luminally -expressed membrane transport mechanisms such as $Na^+$-coupled bicarbonate transporters and the $Na^+/K^+$-ATPase also contribute to CSF production[4,13,14], and may potentially cotransport water. Figure 7b illustrates a summary of the coupled transporters expressed in the choroid plexus[14,54]. The involvement of bicarbonate transporters has been investigated via inhibition of the carbonic anhydrase by acetazolamide[55,56]. As this treatment appears highly vasoconstrictive especially in choroid plexus[57], the ensuing decreased choroidal blood flow, which in itself lowers CSF production[38], may complicate delineation of the direct role of the bicarbonate transporters. The slightly increased effect of furosemide over that of bumetanide in the ventricular-cisternal perfusion experiment, despite the lack of KCCs in the luminal membrane of the choroid plexus, may be assigned to the low-affinity inhibitory action of furosemide on carbonic anhydrase[52] and potentially on chloride/bicarbonate exchange[58]. The less invasive and rapid whole animal imaging qualitatively confirmed the role of NKCC1 in CSF production.

Notably, genetic deletion of NKCC1 is predicted to cause severe alterations in choroidal epithelium ion concentrations, and

therefore in the driving forces and activity of other choroidal transport mechanisms (which could add a substantial confounding element to data obtained in the NKCC1$^{-/-}$ mouse model). While NKCC1$^{-/-}$ mice are viable, they, in addition, display a range of physiological deficits[59] and we therefore relied on the well-tested action of bumetanide. With such a specific and effective inhibitor, the advantage of the pharmacological approach is the acuteness of the transporter inhibition, maintenance of normal epithelial ion concentrations, the ability to employ each mouse as its own control, and the guarantee of no developmental effects (as observed for the NCBE/NBCn2$^{-/-}$ mice, in which the luminally localized $Na^+/H^+$ exchanger (NHE1) trafficked to the opposite choroidal membrane[60]).

In support of the proposed importance of NKCC1 in CSF production, a recent report convincingly demonstrated hyperactivation of NKCC1 by inflammatory markers in the CSF in an animal model of intraventricular hemorrhage[9]. This condition is well-known to promote posthemorrhagic hydrocephalus in patients[61], and the increased NKCC1 activity in the animal model yielded bumetanide-sensitive ventriculomegaly. NKCC1, via its direct contribution to brain water accumulation, thus represents a promising pharmacological target in brain pathologies involving disturbed water dynamics and increased intracranial pressure, the lack of which precludes efficient pharmacological treatment of a range of patients.

Taken together, our results challenge the general conception that simple, passive movement of water suffices to support the high CSF production rate. NKCC1-mediated cotransport of water ultimately provides a molecular mechanism by which the CSF production can take place independently of an osmotic gradient in a manner dictated by the prevailing ion gradients, generated and maintained by the concerted action of the $Na^+/K^+$-ATPase and the wealth of other ion transporters and channels expressed in choroid plexus. While the present study addressed the molecular mechanisms of fluid flow from the choroidal epithelial cell to the ventricle, future studies must delineate the transport mechanisms underlying ion and fluid flow across the basolateral membrane. Of immediate interest in this context is the orchestrated synergy between several bicarbonate transporters[14], the expression of which is illustrated in Fig. 7b.

## Methods

**Experimental animals**. All procedures involving animal experimentation conformed to European guidelines, complied with all relevant ethical regulations, and were approved by the Danish Animal Experiments Inspectorate with permission no. 2016-15-0201-00944. Adult male B6JBOM (Taconic) or C57BL6J (Janvier Labs) mice were used for the animal experimentation in ages ranging from 8 to 12 weeks. Housed with 12:12 light cycle and access to water and food ad libitum. Sample size for the in vivo studies was chosen using the formula[62]: sample size = $2SD^2(1.96 + 0.842)^2/d^2$ with 80% power and type 1 error and using SD (standard deviation) = 0.06 from ref. [15] and $d$ (effect size) = 0.11 combined from refs. [15,51].

**Isolation of choroid plexus from mouse brain**. Choroid plexus (CP) was isolated after cervical dislocation, decapitation and rapid removal and placement of the brain in cold artificial CSF solution (aCSF-HEPES) containing (in mM: 120 NaCl, 2.5 KCl, 2.5 CaCl$_2$, 1.3 MgSO$_4$, 1 NaH$_2$PO$_4$, 10 glucose, 17 Na-HEPES, pH 7.4). The brains were kept in ice-cold aCSF-HEPES for a minimum of 10 min to cool the tissue and ease dissection. Choroid plexus was isolated from the lateral and the fourth ventricles (the latter only employed in expression studies and ex vivo volume live imaging) after removal of the most lateral 2−3 mm of the brain hemispheres followed by separation of the two brain hemispheres.

**Live imaging**. Following isolation, the choroid plexus was stored in ice-cold aCSF-HEPES up to 3 h prior to experiments and mounted on glass coverslips coated with Cell-Tak® (BD Biosciences), prepared according to the manufacturer's instructions (1 part 2 M Na$_2$CO$_3$:9 parts Cell-Tak, addition of 2% isopropanol to decrease surface tension, washed twice in aCSF after 30 min drying period). The coverslip was placed in the closed laminar perfusion chamber (Warner instruments) prior to loading with 16.67 μM calcein-AM (Life tech), 8–10 min loading followed by a 6–7 min washout period to remove excess dye. Live imaging was performed at 37 °C on

choroid plexus mounted on an inverted Nikon T2000 microscope stage with a ×60 1.4 NA plan Fluor objective placed underneath the coverslip with immersion oil. The tissue was superfused with aCSF-HEPES at a flow rate of 1 ml min$^{-1}$ and swift solution change was ensured with a PC-16 controller (Bioscience Tools). Hyperosmolar solutions were obtained by addition of 100 mOsm mannitol (100 mM), NaCl (55 mM), or KCl (55 mM). Na$^+$-free aCSF-HEPES (in mM): 120 cholineCl, 2.5 KCl, 2.5 CaCl$_2$, 1.3 MgSO$_4$, KH$_2$PO$_4$, 17 HEPES, 10 glucose (15 mannitol to obtain ~290 mOsm), pH 7.4). Osmolarities were determined with an accuracy of 1 mOsm with an osmometer Type 15 (Löser Messtechnik). The fluorophore was excited by light of a wavelength of 495 ± 15 nm delivered by a polychrome IV monochromator (Photonics). The emitted fluorescence of wave lengths 510–535 nm was recorded for 2–10 ms at a frequency of 1–5 s with a 12-bit cooled monochrome CCD (Charge-Coupled Device) camera and analyzed with FEI Live Acquisition software. Each choroid plexus was randomly assigned and all image analyses were blinded to the experimental conditions. Images of mouse choroid plexus were converted into black, and background into white using ImageJ percentile threshold adjustment. The 2D changes in the black to white ratio were used to measure relative changes in the size of the choroid plexus. These values represent an underestimation of the actual 3D volume changes, assuming these are approximately isotropic but with the relative comparisons employed in this study, this limitation does not affect the outcome of our data.

**mRNA quantification.** Total RNA from mouse choroid plexus in RNAlater (R0901, Sigma-Aldrich) was purified with the RNeasy micro kit (Qiagen) and the RNase-free DNase kit (Qiagen), according to the manufacturer's instruction. A total of 0.5 µg total RNA or 1 µg cRNA was used for reverse transcription using the Omniscript RT mini kit (Qiagen) (cDNA from cRNA was diluted 1:50,000 before further use). cDNA was amplified by quantitative PCR conducted using LightCycler 480 SYBR Green I Master Mix (Roche Applied Sciences). Reactions were carried out in triplicates on a Stratagene Mx3005P QPCR system from Agilent Technologies. The following primer pairs were applied for amplification of targets: 5′-GCAAGACTCCAACTCAGCCAC-3′ (forward) and 5′-ACCTCCATCATC AAAAAGCCACC-3′ (reverse) to generate an *SLC12A2* product of 158 bp; 5′-GCCCCAACCTTACTGCTGAC-3′ (forward) and 5′-TCTCCTTTAGGCCGA GGGTG -3′ (reverse) to generate an *SLC12A4* product of 150 bp; 5′-TGCTCA TTGCCGGGTCATT-3′ (forward) and 5′-CCACGTTCTGATCCTGGTCC-3′ (reverse) to generate an *SLC12A5* product of 195 bp; 5′-CAGCTGGGGGCTCATA CTTC-3′ (forward) and 5′-ACTCCGAAAGATGGCAGCTC-3′ (reverse) to generate an *SLC12A6* product of 170 bp; 5′-AGCTCAACGGCGTAGTTCTC-3′ (forward) and 5′-CTGTTCAGCCCTTCCGTCAG-3′ (reverse) to generate an *SLC12A7* product of 136 bp; 5′-CCGGGTTCCTATAAATACGGACTG-3′ (forward) and 5′-CAATCTCCACTTTGCCACTGC-3′ (reverse) to generate a Glyceraldehyd-3-phosphate (*GAPDH*) product of 195 bp; 5′-CTCCGGAAAGG CCAAGACAA-3′ (forward) and 5′-TTTGACGCATTTCCTGCCAAC-3′ (reverse) to generate an H2A family, member Z (*H2AFZ*) product of 198 bp.

Primers were designed using NCBI's pick primer software. The optimum concentration for each primer set was determined to 200 nM. The initial melting was performed at 95 °C for 10 min. During the subsequent 40 amplification cycles, the melting temperature was 95 °C (20 s), the primer annealing temperature was 60 °C (22 s), and the elongation temperature was 72 °C (20 s). After completed amplification, melting curves were generated to confirm amplification specificity. Standard curves of 4× serial dilutions of cDNA were made using either reverse transcribed cRNA (target genes) or total RNA from mouse choroid plexus (reference genes) in order to determine the amplification efficiencies for each of the utilized primer-sets (*SLC12A2* 104.3%; *SLC12A4* 95.1%; *SLC12A5* 101.2%; *SLC12A6* 101.8%; *SLC12A7* 96.1%, *GAPDH* 100.5%; and *H2AFZ* 97.6%). GenEx (MultiD Analyses AB) was used for data analysis including testing for best reference gene combinations. Target genes are normalized to *GAPDH* and *H2AFZT* and presented as relative expression.

**Protein expression in oocytes and membrane preparations.** Oocytes from *Xenopus laevis* were obtained from frogs purchased from Nasco (Fort Atkinson) or purchased from Ecocyte Bioscience (Germany). The oocytes were surgically removed from anesthetized (2 g l$^{-1}$ Tricain, 3-aminobenzoic acid ethyl ester, Sigma-Aldrich A-5040) frogs. The follicular membrane was removed by incubation in Kulori medium (90 mM NaCl, 1 mM KCl, 1 mM CaCl$_2$, 1 mM MgCl$_2$, 5 mM HEPES, pH 7.4, 182 mOsm) containing 10 mg ml$^{-1}$ collagenase (type 1, Worthington, NJ, USA) and trypsin inhibitor (1 mg ml$^{-1}$, Sigma-Aldrich, Denmark) for 1 h, prior to wash in Kulori medium containing 0.1% bovine serum albumin (Sigma, Denmark) and incubated in 100 mM K$_2$HPO$_4$ with 0.1% BSA for 1 h. The oocytes were kept in Kulori medium until experiments. cDNA encoding rNKCC1 was obtained from Professor Kai Kaila (University of Helsinki, Finland) and cDNA encoding mKCC1-4 was obtained from Professor Hans Gerd NothWang (Carl von Ossietzky University of Oldenburg, Germany) and all sequences were verified prior to use (Eurofins Genomics). cRNA was prepared from linearized plasmids using the mMESSAGE mMACHINE T7 kit (Ambion) and extracted with MEGAclear (Ambion), according to the manufacturer's instructions, prior to microinjection of 50 ng cRNA per oocyte with a Nanoject microinjector (Drummond Scientific Company). The oocytes were kept at 19 °C for 5 days after which membrane preparations were obtained by homogenization of 20–40 oocytes expressing each

construct in 1 ml buffer containing (in mM: 5 MgCl$_2$, 5 NaH$_2$PO$_4$, 1 EDTA, 80 sucrose, 20 Tris, pH 7.48, containing the protease inhibitors leupeptin (8 µM) and pefabloc (0.4 mM), both from Sigma-Aldrich). The supernatant was recovered following 10 min centrifugation at 250 × $g$ and subsequently centrifuged at 14,000 × $g$ for 20 min to obtain the total membrane fraction.

**Western blotting.** Isolated choroid plexuses (laterals and fourth) were pooled from each mouse and sonicated on ice (3 × 10 s at 70%, Sonopuls, Bandelin) in phosphate-buffered saline (PBS). Western blotting was performed using precast SDS-PAGE gels (Mini-PROTEAN, Biorad) and immobilon FL-membranes (Merck Millipore). Both primary and secondary antibodies were diluted in Odyssey blocking buffer (LI-COR Biosciences):PBS-T at 1:1. Signals were detected with an Odyssey CLx imaging system and image analysis was performed using Image Studio 5 (both from LI-COR Biosciences). Full blots with original markers are included as Supplementary Fig. 1. Primary antibodies: anti-β-tubulin; MAB3408 (Millipore, 1:500), anti-e-cadherin; 610181 (BD BioSciences, 1:5000), anti-NKCC1; sc-21545 (Santa Cruz, 1:500), anti-KCC1&3 [63–66] (kind gift from Professor Thomas J. Jentsch, Max Delbrück Center For Molecular Medicine, Berlin, Germany, 1:500), anti-KCC2; 07-432 (Millipore, 1:1000), anti-KCC3&4 [67] (kind gift from Professor Jinwei Zhang, University of Dundee, UK, 2 µg ml$^{-1}$), other anti-KCC3 tested; HPA034563 (Atlas, 1:200), H00009990-A01 (Abnova, 1:1000), and sc-19424 (Santa Cruz, 1:200). Secondary antibodies; IRDye 800CW donkey anti-goat; P/N 926-32214, IRDye 680RD donkey anti-mouse; P/N 926-68072, IRDye 800CW donkey anti-rabbit; P/N 926-32213, and IRDye 800CW goat anti-rabbit; P/N 926-32211. All from LI-COR Biosciences, 1:10,000-1:15,000.

**Cell surface biotinylation.** Mice were anesthetized with isoflurane, placed in the prone position in a stereotaxic frame (Harvard Apparatus), a dorsal midline incision was made over the skull and upper cervical spine to expose the cranium, after which a brain infusion cannula (brain infusion kit 3, Alzet) was placed in the lateral ventricle using the coordinates: 1.0 mm lateral to the midline, 0.5 mm posterior to bregma, and 2.5 mm ventral into the brain. After sacrificing the mouse by cervical dislocation, 1.5 mg EZ-link Sulfo-NHS-SS biotin (Thermo Fisher) in 100 µl biotin buffer (in mM: 125 NaCl, 2 CaCl$_2$, 10 triethanolamine, pH 7.5) was injected to the ventricle and the mouse left on ice for 15 min followed by isolation of choroid plexus. After isolation and quenching to remove excess biotin, the choroid plexus was transferred to a lysis buffer (in mM: 150 NaCl, 5 EDTA, 50 Tris-HCl, 1% Triton X-100, 0.05% SDS, 0.4 pefabloc and 8 µM leupeptin) for 30 min, all according to the manufacturer's instructions. The samples were sonicated 3 × 10 s at 70% (Sonopuls, Bandelin) and centrifuged at 10,000 × $g$ for 5 min at 4 °C. An aliquot was removed (total fraction) before proceeding with the biotin (luminal) fraction purified on NeutrAvidin (Thermo Fisher) columns (Pierce).

**Tissue preparation and immunostaining.** Sections obtained from tissue from male C57BL/b mice from Taconic (Denmark) were immunostained with anti-NKCC1(kind gift from Professor Turner[68], 1:4000) and anti-e-cadherin; 610181 (BD Biosciences, 1:2000), prior to labeling with secondary antibody; donkey anti-mouse; Alexa555 (Invitrogen, 1:1000), donkey anti-rabbit; Alexa488 (Invitrogen, 1:1000) and nuclear staining with Topro3 (Invitrogen, 1:1000). Anti-KCC1 tested; HPA041138 (Atlas Antibodies), anti-KCC1 (kind gift from Professor Thomas J. Jentsch), and ab115607 (Abcam). The digital images were acquired with a DM IRE2 inverted confocal microscope (Leica Microsystems).

**Ion content determination.** To obtain enough material, samples from four mice were pooled for each of the four experiments. Mice were anesthetized with xylazine (only one initial intraperitoneal (i.p.) injection; 1 mg ml$^{-1}$ and 0.1 ml per 10 g body weight, 37 °C, ScanVet). 5−10 min later, it was followed by an i.p. injection with ketamine (10 mg ml$^{-1}$ and 0.1 ml per 10 g body weight + 150 µl for mice <30 g and 200 µl for mice >30 g, 37 °C, MDS Animal Health). With intervals of 25−35 min, the mouse was re-dosed with ketamine (up to 50% of start dose, as needed to sustain anesthesia). Anesthetized mice, after tracheotomy, were placed in a stereotaxic frame and clear CSF collected from cisterna magna. Immediately after decapitation, blood was collected and choroid plexus isolated, weighed, dried at 105 °C, and extracted in 0.1 M HNO$_3$ on a horizontal shaker table (200 rpm, 72 h, room temperature)[69]. The ion concentrations of the choroid plexus were calculated based on previous observations of choroid plexus epithelium containing 79% water and 8.5% blood[69]. Na$^+$ and K$^+$ content was quantified by flame photometry (Instrument Laboratory 943) while Cl$^-$ concentration was quantified by a colorimetric method using QuantiChrom™ Chloride Assay Kit (MEDIBENA Life Science & Diagnostic Solution). Gibbs free energy for NKCC1 was calculated as Eq. 1

$$\Delta G = RT \times \ln \frac{[Na^+]_i \times [K^+]_i \times ([Cl^-]_i)^2}{[Na^+]_o \times [K^+]_o \times ([Cl^-]_o)^2},\qquad(1)$$

where $R$ = gas constant (8.314 J mol$^{-1}$), $T$ = temperature (at 37 °C; 310.15 K), $[X]_i$ = intracellular ion concentration, and $[X]_o$ = ion concentration in CSF.

**Imaging of intracellular sodium and cell viability**. For Na$^+$ imaging of choroid plexus bathed in aCSF-HEPES, the epithelial cells were loaded (~20 min) with the membrane-permeable form of SBFI (sodium-binding benzofuran isophthalate acetoxymethyl (AM) ester, 200 μM, Teflabs) To this end, the dye was pressure-applied (5 s) directly onto the cells on several positions[70]. Afterwards, the tissue was perfused with aCSF-HEPES for at least 20 min to allow for de-esterification of the dye before imaging experiments were commenced. Wide-field Na$^+$ imaging was performed utilizing a variable scan digital imaging system (Nikon NIS-Elements v4.3, Nikon) attached to an upright microscope (Nikon Eclipse FN-PT, Nikon GmbH). The microscope was equipped with a ×40/N.A. 0.8 LUMPlanFl water immersion objective (Olympus Deutschland GmbH) and an orca FLASH V2 camera (Hamamatsu Photonics Deutschland GmbH). SBFI was alternatively excited at 340 and 380 nm. Images were obtained at 1 Hz and emission was collected >440 nm. Fluorescence was evaluated in regions of interest (ROIs), representing single cells. Signals were background-corrected as follows: The fluorescence intensities obtained at each excitation wavelength (340/380 nm) were dynamically subtracted frame by frame by the respective fluorescence emissions of a ROI in the field of view, which was free of SBFI-labeling (background). After background subtraction, the fluorescence ratio (F$_{340}$/F$_{380}$) was calculated from individual ROIs and analyzed using OriginPro Software (OriginLab Corporation v.9.0). For imaging of cell survival of the choroid plexus after bumetanide treatment, the epithelial cells were loaded with Calcein-AM (500 μM, Sigma-Aldrich) as described for SBFI loading. This was done after 10 min of bumetanide treatment, right before the blocker washout (or after 10 min of aCSF as a control). Calcein was excited at 488 nm, emission was collected 510 nm. Each choroid plexus was randomly assigned to each group.

**$^{86}$Rb$^+$ efflux experiments**. Choroid plexus was isolated as above, initially in cold aCSF (in mM: 120 NaCl, 2.5 KCl, 2.5 CaCl$_2$, 1.3 MgSO$_4$, 1 NaH$_2$PO$_4$, 25 NaHCO$_3$, 10 glucose, pH 7.4, equilibrated with 95% O$_2$/5% CO$_2$) but allowed to recover at 37 °C for 5–10 min before beginning of the experiment. Choroidal isotope accumulation was performed by a 10 min incubation in equilibrated (95% O$_2$/5% CO$_2$) aCSF-based isotope medium (2 μCi ml$^{-1}$ $^{86}$Rb$^+$, NEZ07200 (congener for K$^+$ transport) and 8 μCi ml$^{-1}$ $^3$H-mannitol, NET101 (extracellular marker), both from PerkinElmer), followed by 15 s wash prior to incubation in 0.5 ml equilibrated (95% O$_2$/5% CO$_2$) efflux medium (aCSF containing 20 μM bumetanide, 1 mM furosemide or vehicle (DMSO), each choroid plexus randomly assigned to each group). 0.2 ml of the efflux medium was collected into scintillation vials every 20 s (time points: 0, 20, and 40 s) and replaced with fresh aCSF. At the end of the experiment, choroid plexus was solubilized at room temperature with 1 ml Solvable (6NE9100, PerkinElmer) in the leftover efflux medium. The isotope content was determined by liquid scintillation counting with Ultima Gold$^{TM}$ XR scintillation liquid (6013119, PerkinElmer) in a Tri-Carb 2900TR Liquid Scintillation Analyzer (Packard). The choroid plexus $^{86}$Rb$^+$ content corrected for $^3$H-mannitol (extracellular background) was calculated for each time point, and the natural logarithm of the choroid plexus content $A_t$/$A_0$ was plotted against time[47]. Slopes indicating the $^{86}$Rb$^+$ efflux rate constants (s$^{-1}$) were determined from linear regression analysis.

**Ventriculo-cisternal perfusion**. Mice were anaesthetized with ketamine and xylazine (as described above) during the experimental procedure and their body temperature monitored and maintained at approximately 37 °C using a home-othermic system (Harvard Apparatus). To obtain near-physiological conditions, the mice were ventilated (VentElite, Harvard Apparatus) after tracheotomy and settings continuously optimized for each animal using a capnograph (Type 340, Harvard Apparatus) and a pulse oximeter (MouseOx Plus, Starr Life Sciences), each calibrated with respiratory partial pressure of carbon dioxide and arterial oxygen saturation (ABL800 FLEX, Radiometer). Ventilation of a ~25 g mouse; approximately 150 μl per breath of a mix of 0.1 l min$^{-1}$ O$_2$ and 0.9 l min$^{-1}$ air and approximately 150 breath per min, sight = 10 % increase in tidal volume and Positive End-Expiratory Pressure (PEEP) = 2 cm H$_2$O. The heart rate was continuously monitored (MouseOx Plus, Starr Life Sciences). After tracheotomy, 1 ml of warmed 0.9% NaCl was injected subcutaneously to decrease risk of dehydration. The infusion cannula was placed as described in the biotinylation method, and was glued to the scull (with superglue, Pelikan). The perfusion solution was heated to 37 °C in an inline heater (SF-28, Warner Instruments) before entering the infusion cannula. A micropipette held in a 5° position was introduced into the cisterna magna. After observing CSF in the micropipette, infusion of equilibrated (95% O$_2$/5% CO$_2$) aCSF (containing 1−2 mg ml$^{-1}$ tetramethylrhodamine isothiocyanate-dextran (TRITC-dextran MW 155,000, T1287 Sigma-Aldrich) or 0.03 % w/v Evans blue (314-13-6 Sigma-Aldrich)) and 0.1% DMSO was initiated at a rate of ~ 0.7 μl min$^{-1}$. Addition of dextran and vehicle (DMSO) increased the osmolarity of the aCSF by approximately 20 mOsm. The final osmolarity of the aCSF employed for the ventricular-cisternal perfusion thus amounted to approximately 313 mOsm, designed to match the mouse plasma osmolarity (313 ± 2 mOsm, $n$ = 5 mice) in order to limit osmotically induced experimental confounders into the experimental paradigm. An extra set of mice were anaesthetized with isoflurane (0.5–1.5 %) at the time of CSF production rate determination. Anesthesia was induced as above (to facility the tracheometry) but the animal transferred to isoflurane immediately thereafter with no subsequent additional ketamine injections. In this manner, the

animal had not been subjected to ketamine/xylazine for at least 2 h and the dominant anesthesia predicted to be isoflurane. After an hour of infusion, the animals anesthetized with the standard ketamine/xylazine described above were exposed to a new aCSF solution containing either 2 mM furosemide, 100 μM bumetanide (expected ventricular concentrations of 1 mM and 50 μM) or the vehicle, DMSO (animals were randomly assigned to these groups). The solution change took 5–10 min. Outflow was collected in 5-min intervals by introducing a second micropipette into the fixed cisterna magna micropipette and the fluorescent content measured in a microplate photometer (Fluoroskan Ascent, Thermo Lab-systems). The production rate of CSF was calculated from the equation:

$$V_p = r_i \times \frac{C_i - C_o}{C_o}, \qquad (2)$$

where $V_p$ = CSF production rate (μl min$^{-1}$), $r_i$ = infusion rate (μl min$^{-1}$), $C_i$ = fluorescence of inflow solution, $C_o$ = fluorescence of outflow solution.

**In vivo live imaging**. Ventricular injections were done as above, except with variation in the ventricular depth: A cannula was placed 2.5 mm into the brain and 2 μl aCSF with vehicle(DMSO)/inhibitor (2 mM furosemide or 100 μM bumetanide, animals were randomly assigned to these groups) was injected during 2 s. Within 5 min, the cannula was removed and a new placed 2.2 mm into the brain and the second solution injected. This second cannula contained aCSF with vehicle/inhibitor, as above, in addition to a carboxylate dye (10 μM IRDye 800CW, P/N 929-08972, LI-COR Biosciences). With a lapse of 1 min after injection, the first image was obtained on the Pearl Trilogy Small Animal Imaging System (800 nm channel, 85 μm resolution, and 30 s intervals, LI-COR Biosciences). At the termination of each experiment, a white field image was taken. Due to the swift protocol, these mice were not ventilated. Images were analyzed using Image Studio 5.2 and the region of interest defined as a square, starting at lambda. For test of whether the injected solution reached both ventricles, 0.03 % w/v Evans blue (314-13-6, Sigma-Aldrich) dissolved in aCSF was injected in a manner mimicking the injection of fluorescent dye.

**Chemicals**. Chemicals were freshly dissolved prior to each experimental day. Furosemide (F4381, Sigma-Aldrich) was dissolved directly into the aCSF while a stock solution (100 mM in DMSO) was prepared with bumetanide (B3023, Sigma-Aldrich).

**Statistics**. Data analysis and statistical tests were carried out with GraphPrism 7.0 (GraphPad Software). One-sample $t$ test, one-way or two-way (RM)ANOVA followed by the Tukey's multiple comparisons post hoc test were employed as indicated in figure legends. Data are obtained from choroid plexus from individual animals and presented as mean ± SEM with significance set at $P$ < 0.05.

**Data availability**. We confirm that all relevant data from this study are available from the corresponding author upon request.

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

## Acknowledgements

We thank Mette Assentoft, Micael Loenstrup, Kristofffer Racz, and Carina Lynnerup Pedersen for expert technical assistance and Professor Jeppe Praetorius, Aarhus University, Denmark for providing the immunostaining of NKCC1 in choroid plexus. We are grateful to Professor Thomas J. Jentsch, University of Hamburg, Germany for kindly providing the KCC1 and KCC3 antibodies and Professor Jinwei Zhang, University of Dundee, Scotland for kindly providing the KCC3 and KCC4 antibodies. We thank Professor Thomas Zeuthen for critical reading of the manuscript. This work was supported by Thorberg's Foundation (53.734), Novo Nordisk Foundation; Tandem program (NNF17OC0024718), the Danish Medical Research Council, Sapere Aude program (0602-02344B), Vera and Carl Johan Michaelsen's Scholarship, the Augustinus Foundation, Doctor Sofus Carl Emil Friis og hustru Olga Doris Friis' scholarship, and the Carlsberg Foundation (CF15-0070).

## Author contributions

A.B.S., K.T., C.R.R., and N.M. designed the research; A.B.S., E.K.O, A.S., N.J.G., D.B., K. T., and C.R.R. performed research and analyzed the data; A.B.S and N.M. drafted the manuscript and all authors contributed to the final version of the manuscript.

## Additional information

**Competing interests:** The authors declare no competing interests.

