## [Peer Review File · Nature Communications]

Reviewers' comments:

Reviewer #1 (Remarks to the Author):

This interesting manuscript describes the role of choroidal NKCC1 in spinal fluid production. In general the paper is well-written, the experimental logic is sound, and the data are solid. Comments:

The source of CSF production has been investigated for a long time. Recent reviews have emphasized other transporters, for example Na-dependent acid-base transporters (e.g. Christensen et al. 2013). Figure 7 does not provide a sense of the number of transporters that have been described in the choroid plexus. While the simplicity of Fig 7 is very useful for focusing on the novel findings of this study, the readers of a general publication such as Nat Comm will benefit from a second panel that includes the other verified transporters. This information could also provide another explanation for the limited efficacy of bumetanide in the inhibition of CSF production.

The Gibbs free energy for NKCC1-mediated CSF transport calculated for the 55 mM KCl condition (in which choroid swelled rather than shrank) strongly favors transport into the choroid. However, inward transport also seems to be favored in the experiment with 55 mM NaCl added ($K_o = 3$; $N_{ao} = 205$; $Cl_o = 155$; $K_i = 141$, $N_{ai} = 31$; $Cl_i = 35$). Perhaps a table of ion concentrations and the calculated ratios would be helpful, as would an explanation as to why outward transport was still observed after increasing N_{ao} and Cl_o .

Minor:

In the paragraph NKCC1 and KCC are expressed in the choroid plexus, the specificity of the antibodies are tested in a xenopus expression system. This seems to be a test of the sensitivity, not the specificity, of the antibodies. To be a test of specificity would require that xenopus oocytes natively express all the other transporters that could underlie the observed experimental phenomena.

I would hesitate to include idiopathic intracranial hypertension as a disorder of CSF overproduction. Compared to choroid plexus tumors in which CSF overproduction is verified, the ventricular anatomy is completely different in IIH (slit-like ventricles vs enlarged).

I was surprised that IV bumetanide had no effect on CSF production. It is sometimes used clinically to good effect. The explanation, that bumetanide does not reach the luminal side of the choroid plexus epithelia, could be supported with citations or data.

Reviewer #2 (Remarks to the Author):

It is not often to be able to say that it was a pleasure to review a manuscript but it certainly was in this case. The paper is clearly written and provides appropriately detailed information about the methods used.

The authors have studied a decades old problem with expertise and imagination. The source of water for secreted CSF has been a matter of unsatisfactory speculation supported by inadequate mathematical models and experimental data. The authors have used a comprehensive and complementary combination of classical physiological in vivo and ex vivo techniques together with molecular biology, immunohistochemistry and pharmacology. They have shown that The Na⁺/K⁺/2Cl⁻ co-transporter (NKCC1) expressed in the luminal membrane of

choroid plexus is probably responsible for a substantial proportion, perhaps as much as 50%, of CSF production into the ventricular system, with coupled water and ion transport. The K⁺/Cl⁻ co-transporter (KCC1) in the basolateral membrane has a complimentary function in transfer of water and ions from the interstitial space of the choroid plexus into the epithelial cells of the plexus. Of particular importance is the finding that furosemide, which inhibits KCCs and NKCCs blocked the K⁺-mediated choroid plexus increase when challenged with hypertonic mannitol solutions.

It has been unclear for decades how the water component of secreted CSF is transferred to the ventricular system. With discovery of the high concentration of the water channel aquaporin-1 (AQP-1) in the apical membrane of the choroid plexus epithelial cells it has been assumed that this would be important in the process. However, the sparse or as claimed by some (e.g. Nielsen et al. 1993. Proc Natl Acad Sci USA 90, 7275–7279) absence in the basolateral membrane, left unanswered the question of the mechanism by which water entered the epithelial cells from the interstitial space of the plexus. The authors cite the finding that in AQP-1 null mice CSF secretion is reduced by only 20% as evidence that this is not an important mechanism contributing to CSF secretion. However, knock-out experiments are difficult to interpret in functional terms as it is unknown what compensatory mechanisms may have occurred during development. In that context it would be interesting to know if there is any up regulation of NKCC1 and KCC1 in these mice.

My only question is what was the basis for the composition of the artificial CSF used? The NaCl concentration in particular seems rather low (cf Davson & Segal, 1996, Physiology of the CSF and Blood-Brain Barriers, CRC Press, Boca Raton pp16-18) and their own results on page 6. Thus the osmolality of the aCSF may have been rather low. Also it is generally preferable to express units for CSF (and plasma) ionic composition as /Kg or /L of H₂O to take account of differences in protein concentration (Davson & Segal, 1996) between different fluids and physiological/pathological conditions. Was this the composition of the fluid used for ventriculo-cisternal perfusion. It is a considerable technical feat to have carried this out mice, which have only a small ventricular system. It is also noteworthy that considerable attention was paid to maintain the animals in a "normal" physiological state. It has been unclear how the water component of secreted CSF is transferred to the ventricular system.

Reviewer #3 (Remarks to the Author):

Steffensen et al., investigate water transport required for CSF production using an ex vivo choroid plexus preparation and in vivo in mice. According to the traditional view of CSF

production, water required for CSF production passes from the blood to the CSF via aquaporin channels present on the apical membrane of choroid plexus epithelial cells. This model is at odds with the results from studies performed in knockout mice for example, which suggest additional mechanisms contribute to water transport across this barrier. The present study suggests the exciting possibility that water co-transport through NKCC1 allows water to move against an osmotic gradient and that NKCC1 provides a previously unappreciated means by which water is delivered across the epithelium for CSF production. Several questions remain:

1. As sleep and anesthesia can alter fluid flow in the brain (see work from Nedergaard group, some of which is referenced in manuscript), it would be better to analyze CSF ion concentrations from cannulated mice that are awake and behaving, and/or from mice naturally asleep, and/or compare between mice exposed to different anesthetics to expose potential variations in proposed CSF water transport model compared to naturalistic state. As heart rate and respiration can also influence CSF production, this point is worth careful consideration (see for example Hart et al., *Am J Physiol Heart Circ Physiol*, 2001).
2. Figures 2-3 focus on validating antibodies and expression localization. It seems that many antibodies have already been published and used by others, which would suggest that these data should be in supplemental figures. However as presented, it is unclear what these data contribute to the present study. Review articles with extensive figures/schematics clearly explaining transporter localization have been published several times by experts in the field (e.g. reviews from Praetorius group; Damkier et al.). The authors also use choroid plexus tissues from different ventricles in different experiments (e.g. pooling lateral and 4th for westerns), but recent findings have shown that lateral and 4th ventricle choroid plexus have distinct transcriptomes and gene expression patterns can change during lifespan (e.g. Lun et al., *J Neurosci*, 2015).
3. Fig. 4B –How can the statement that the baseline was “rather stable” be made from a 1 minute baseline calculation? Upon inhibition with bumetanide, the authors observe changes in SBF1 fluorescence ratios. Are the cells similarly healthy based on calcein-AM signal at the end of each experiment? Are the effects of inhibitors reversible during a washout? How long does it take to return to baseline? This point is pertinent for all inhibitors used.

We thank the reviewers for their positive comments and thoughtful and helpful suggestions. We have in the revised version of the manuscript included a new set of experiments on CSF production in mice with an alternative anesthesia and included further experiments on live Na⁺ imaging with inhibitor wash-out and cell viability determination. We have provided point-to-point answers to the reviewer comments (below, marked with blue) and marked all changed in the revised manuscript (marked as such) in red font (highlighted a few lines already in the earlier version in green). We feel that the suggested alterations have clearly improved the manuscript and we hope that we were able to satisfy the reviewers' concerns.

Reviewers' comments:

Reviewer #1 (Remarks to the Author):

This interesting manuscript describes the role of choroidal NKCC1 in spinal fluid production. In general the paper is well-written, the experimental logic is sound, and the data are solid. Comments:

The source of CSF production has been investigated for a long time. Recent reviews have emphasized other transporters, for example Na-dependent acid-base transporters (e.g. Christensen et al. 2013). Figure 7 does not provide a sense of the number of transporters that have been described in the choroid plexus. While the simplicity of Fig 7 is very useful for focusing on the novel findings of this study, the readers of a general publication such as Nat Comm will benefit from a second panel that includes the other verified transporters. This information could also provide another explanation for the limited efficacy of bumetanide in the inhibition of CSF production.

Answer: We thank the reviewer for the suggestion and have included a second panel in Figure 7, which includes the other relevant choroidal cotransporters that very likely will contribute to the CSF production as also mentioned in the text on pages 12-13.

The Gibbs free energy for NKCC1-mediated CSF transport calculated for the 55 mM KCl condition (in which choroid swelled rather than shrank) strongly favors transport into the choroid. However, inward transport also seems to be favored in the experiment with 55 mM NaCl added ($K_o = 3$; $N_{ao} = 205$; $Cl_o = 155$; $K_i = 141$, $Na_i = 31$; $Cl_i = 35$). Perhaps a table of ion concentrations and the calculated ratios would be helpful, as would an explanation as to why outward transport was still observed after increasing N_{ao} and Cl_o .

Answer: We calculated the Gibbs free energy for NKCC1 in standard physiological conditions in association with the experiments in Fig. 4 (page 6, in which the ion concentrations are listed, marked in green in the resubmitted version), not for the experimental conditions (of high KCl or high NaCl) illustrated in Fig. 1. In this experimental setting, we only determine volume changes and not rate of transport (as we do in Fig. 4). Still, the reviewer's point is well taken. NKCC1 is likely to transport inwards when faced with high NaCl concentration on the outside. With the high $K_m(K^+)$, which we in astrocytes determined to > 20 mM (Larsen BR et al., GLIA 2014), its inward transport rate will be relatively low with 3 mM K⁺. The associated volume translocation may simply be insignificant compared to the large outwardly-directed osmotic water flux. So in essence, the experimental set-up in Fig. 1 (which the reviewer refers to) does not allow us to directly determine at what rate NKCC1 transports with high NaCl externally but only the resultant volume changes (osmotic versus cotransport).

Minor:

In the paragraph NKCC1 and KCC are expressed in the choroid plexus, the specificity of the antibodies are tested in a *Xenopus* expression system. This seems to be a test of the sensitivity, not the specificity, of the antibodies. To be a test of specificity would require that *xenopus* oocytes natively express all the other transporters that could underlie the observed experimental phenomena.

Answer: We apologized for not being sufficient clear on our approach and the section has now been rewritten for clarity (page 5). Our approach relied on heterologous expression of all the relevant transport proteins individually in *Xenopus* oocytes. In this manner, we could load the gel with lysate containing all the different transporters and then blot each gel with each antibody. So the NKCC1 antibody was applied to lysate containing NKCC1, KCC1, KCC2, KCC3, or KCC4 (in addition to uninjected oocytes that did not express any exogenous transport proteins). On this gel, a band was *only* detected on the lane containing NKCC1 and not the others – illustrating the anti-NKCC1 antibody that we used did not recognize the KCCs. The identical approach was then done with an antibody recognizing KCC1, etc until we had a set of specific antibodies.

I would hesitate to include idiopathic intracranial hypertension as a disorder of CSF overproduction. Compared to choroid plexus tumors in which CSF overproduction is verified, the ventricular anatomy is completely different in IIH (slit-like ventricles vs enlarged).

Answer: We thank the reviewer for the comments and have removed IIH from the introduction. We have in addition added the newly published observation that posthemorrhagic hydrocephalus may occur as a function of CSF hyper secretion (Karimy et al., Nature medicine 2017), page 3.

I was surprised that IV bumetanide had no effect on CSF production. It is sometimes used clinically to good effect. The explanation, that bumetanide does not reach the luminal side of the choroid plexus epithelia, could be supported with citations or data.

Answer: We agree with the reviewer that some CCC inhibitors may occasionally be used in the clinic for elimination of excess brain water fluid. The neurologists at the University Hospital in Copenhagen never do (personal communication with Senior Consultant Christina Kruuse, Department of Neurology, Herlev Hospital, Denmark and Senior Consultant Marianne Juhler, Department of Neurosurgery, Rigshospitalet, Denmark, due to the poor effect they observe). CCC inhibitors will promote systemic water loss, which indirectly will affect brain water content. The lack of systemic effect of bumetanide/furosemide on experimental animals (sometimes nephrectomized to avoid the action of the kidney), have been consistently documented in the literature. We have included further references illustrating this observation in the relevant section in the discussion, page 12.

Reviewer #2 (Remarks to the Author):

It is not often to be able to say that it was a pleasure to review a manuscript but it certainly was in this case. The paper is clearly written and provides appropriately detailed information about the methods used. The authors have studied a decades old problem with expertise and imagination. The source of water for secreted CSF has been a matter of unsatisfactory speculation supported by inadequate mathematical models and experimental data. The authors have used a comprehensive and complementary combination

of classical physiological in vivo and ex vivo techniques together with molecular biology, immunohistochemistry and pharmacology. They have shown that The Na⁺/K⁺/2Cl⁻ co-transporter (NKCC1) expressed in the luminal membrane of choroid plexus is probably responsible for a substantial proportion, perhaps as much as 50%, of CSF production into the ventricular system, with coupled water and ion transport. The K⁺/Cl⁻ co-transporter (KCC1) in the basolateral membrane has a complimentary function in transfer of water and ions from the interstitial space of the choroid plexus into the epithelial cells of the plexus. Of particular importance is the finding that furosemide, which inhibits KCCs and NKCCs blocked the K⁺-mediated choroid plexus increase when challenged with hypertonic mannitol solutions.

It has been unclear for decades how the water component of secreted CSF is transferred to the ventricular system. With discovery of the high concentration of the water channel aquaporin-1 (AQP-1) in the apical membrane of the choroid plexus epithelial cells it has been assumed that this would be important in the process. However, the sparse or as claimed by some (e.g. Nielsen et al. 1993. Proc Natl Acad Sci USA 90, 7275–7279) absence in the basolateral membrane, left unanswered the question of the mechanism by which water entered the epithelial cells from the interstitial space of the plexus. The authors cite the finding that in AQP-1 null mice CSF secretion is reduced by only 20% as evidence that this is not an important mechanism contributing to CSF secretion. However, knock-out experiments are difficult to interpret in functional terms as it is unknown what compensatory mechanisms may have occurred during development. In that context it would be interesting to know if there is any up regulation of NKCC1 and KCC1 in these mice.

Answer: We thank the reviewer for the kind words expressing his/her enthusiasm about our work and for the constructive suggestion.

[redacted]

Upon the suggestion from the reviewer, we contacted our colleagues who breed the mentioned AQP1^{-/-} mice and will include quantification of choroidal transport proteins in these mice versus their wild type counterpart. These mice, however, will not be bred within the next 3-4 months and an additional 9 weeks are required to obtain the age of the mice used for the present study. Thus, results will not be available before October this year. We, therefore, prefer not to include these data in the present manuscript. Instead, we suggest to include them into our separate study, which focuses on quantification of the osmotic influence on CSF production.

My only question is what was the basis for the composition of the artificial CSF used? The NaCl concentration in particular seems rather low (cf Davson & Segal, 1996, Physiology of the CSF and Blood-Brain Barriers, CRC Press, Boca Raton pp16-18) and their own results on page 6. Thus the osmolality of the aCSF may have been rather low. Also it is generally preferable to express units for CSF (and plasma) ionic composition as /Kg or /L of H₂O to take account of differences in protein concentration (Davson & Segal, 1996) between different fluids and physiological/pathological conditions. Was this the composition of the fluid used for ventriculo-cisternal perfusion.

Answer: We apologize for not being sufficiently clear on this point. We had a strong focus on maintaining the osmolarity of the aCSF comparable to that of the mouse plasma. The osmolarity of the mouse plasma amounted to 313 mOsm in our experiments, n=5 (now included in the methods section, page 20). Inclusion of the dextran required for the ventricular-cisternal approach in addition to the inhibitor (or vehicle) increased the osmolarity of the aCSF by around 20 mOsm. The equilibrated (95 % O₂/5 % CO₂) aCSF

employed in the ventriculo-cisternal perfusion had a total concentration of 146 mM Na⁺ and 127,5 mM Cl⁻ (see page 19, marked in green) in line with that measured in rabbit (Na⁺ = 149 mM and Cl⁻ = 130 mM) and dog (Na⁺ = 151 mM and Cl⁻ = 133 mM) CSF in the review from Damkier et al., Phys Rev 2013, calculated from data from Davson and Segal (reviewers ref.). In our study, the osmolarity of the aCSF was ~293 mOsm, which increased to ~313 mOsm after addition of both dextran and vehicle/inhibitor. While we do acknowledge the use of /L or /Kg in the literature, we would – if the reviewer permits – wish to keep it as mM, as for the general reader who is not familiar with the literature, it may be more difficult to relate to numbers given as mEq/Kg.

It is a considerable technical feat to have carried this out in mice, which have only a small ventricular system. It is also noteworthy that considerable attention was paid to maintain the animals in a “normal” physiological state. It has been unclear how the water component of secreted CSF is transferred to the ventricular system.

Answer: Thank you very much. It was indeed a challenge to combine attention to all physiological parameters and carry out the ventricular-cisternal perfusion in mice. But as the reviewer points out (and what we experienced), it is crucial to make this effort to increase the fidelity in the data.

Reviewer #3 (Remarks to the Author):

Steffensen et al., investigate water transport required for CSF production using an ex vivo choroid plexus preparation and in vivo in mice. According to the traditional view of CSF production, water required for CSF production passes from the blood to the CSF via aquaporin channels present on the apical membrane of choroid plexus epithelial cells. This model is at odds with the results from studies performed in knockout mice for example, which suggest additional mechanisms contribute to water transport across this barrier. The present study suggests the exciting possibility that water co-transport through NKCC1 allows water to move against an osmotic gradient and that NKCC1 provides a previously unappreciated means by which water is delivered across the epithelium for CSF production. Several questions remain:

1. As sleep and anesthesia can alter fluid flow in the brain (see work from Nedergaard group, some of which is referenced in manuscript), it would be better to analyze CSF ion concentrations from cannulated mice that are awake and behaving, and/or from mice naturally asleep, and/or compare between mice exposed to different anesthetics to expose potential variations in proposed CSF water transport model compared to naturalistic state. As heart rate and respiration can also influence CSF production, this point is worth careful consideration (see for example Hart et al., Am J Physiol Heart Circ Physiol, 2001).

Answer: We thank the reviewer for the constructive comments. We agree that it is possible that anesthesia and sleep/wake can affect the rate of CSF production (our study predominantly investigates the % contribution of NKCC1). To address the reviewer's comment, we determined the rate of CSF production with an alternative anesthesia paradigm (isoflurane) and included these new data in the revised manuscript. Due to technical difficulties with combining the isoflurane mask and the required space to perform tracheotomy (for the ventilation), we had to initiate the anesthesia with ket/xyl. The mice only received one dose. Approximately 40 min after the ketamine i.p. injection, the mice were switched to isoflurane. At least another 30 min went by before the ventricular perfusion started – and then 60 min until

the CSF production rate was determined. Normally ketamine needs to be re-dosed every 30 min and here we measured the CSF production rate more than 2h after the ketamine injection. Therefore, at the point of determination of CSF production rate, the main anesthesia is isoflurane. The value for CSF production rate was similar to that obtained with ket/xyl (compare 0.66 ± 0.02 $\mu\text{l}/\text{min}$ with ket/xyl versus 0.60 ± 0.02 $\mu\text{l}/\text{min}$ with isoflurane). This new data set is now included in the revised manuscript pages 7 and 20.

[redacted]

In order to measure CSF production in awake animals, prior cannulation of the lateral ventricle and the cisterna magna (inserted under anesthesia) will be a requirement. We have deemed the mice too small an animal for this type of investigations and are currently in the process of transferring the technical approach to rats in order to address the point raised by the reviewer. It will be many months before this experimental technique and the ethical approval are established and the question will be addressed in another animal model. We therefore would prefer to not include these experiments in the present manuscript.

We agree with the reviewer that heart rate and respiration (in addition to blood gases, pH, temperature, blood pressure) will have a significant effect on the rate of CSF production. We, according to the reviewer's recommendations, therefore monitor heart rate, control the respiration, monitor blood gases, etc (see the methods section page 19) continuously during the experimental procedure. These remain stable during the experimental period. The description of heart rate monitoring was regrettably hidden in the original submission since the data is obtained by the MouseOx, but is now clarified in the revised version on page 19.

2. Figures 2-3 focus on validating antibodies and expression localization. It seems that many antibodies have already been published and used by others, which would suggest that these data should be in supplemental figures. However as presented, it is unclear what these data contribute to the present study. Review articles with extensive figures/schematics clearly explaining transporter localization have been published several times by experts in the field (e.g. reviews from Praetorius group; Damkier et al.). The authors also use choroid plexus tissues from different ventricles in different experiments (e.g. pooling lateral and 4th for westerns), but recent findings have shown that lateral and 4th ventricle choroid plexus have distinct transcriptomes and gene expression patterns can change during lifespan (e.g. Lun et al., *J Neurosci*, 2015).

Answer: We agree with the reviewer that many excellent schematics are included in the different reviews (many cited in the manuscript). However, some of the transporter localizations have remained disputed or unresolved (as KCC3 and KCC4, for example), probably in part due to lack of verification for isoform specificity. During our quest to obtain isoform-specific antibodies, we tested a battery of the published antibodies, many of which appeared to lack the isoform specificity required for our determinations. We therefore deem it crucial, in this and other studies, to carefully assess the antibodies in question prior to assignment of isoform distribution. Since this panel is rather small (Fig. 2B), but crucial to illustrate the validity of our conclusions, we would prefer if the reviewer would allow us to keep it in the manuscript? To obtain sufficient tissue from each mouse to avoid pooling of choroid plexus from several mice, we used tissue from both the 4th and the lateral ventricles. We thank the reviewer for pointing out the distinct expression profile of the different choroid plexuses and, in retrospect, it would have been beneficial to make every effort to avoid mixing those two types of choroid plexus. We have added a comment to this effect in the discussion (page 11).

3. Fig. 4B –How can the statement that the baseline was “rather stable” be made from a 1 minute baseline calculation? Upon inhibition with bumetanide, the authors observe changes in SBFI fluorescence ratios. Are the cells similarly healthy based on calcein-AM signal at the end of each experiment? Are the effects of inhibitors reversible during a washout? How long does it take to return to baseline? This point is pertinent for all inhibitors used.

Answer: We apologize for not having employed sufficient clear figures. In the revised version of the manuscript, we have included new data in a new panel depicting a combined version of the former panel B (3 min baseline) and former panel C (1 min baseline followed by incubation with bumetanide). This new panel B illustrates the inclusion of bumetanide with a long preceding baseline of 10 min, followed by 10 min exposure to bumetanide, and a 20 min wash-out period, as requested by the reviewer. These data indicate that bumetanide inhibition is reversible and that cell viability is not compromised under these conditions. In addition, illustration of cell viability following bumetanide treatment is carried out with calcein-AM uptake (as suggested by the reviewer, now included as panel D in Fig. 4), see page 7 for description. While this experimental approach lends itself to demonstration of inhibitor washout, we regret that this was not possible in the other experimental paradigms (as far as we can deduce). Bumetanide has been used *in vivo* and *in vitro* for a wealth of different experiments (and as a diuretic in humans) throughout the years and we are of the impression that the cells tolerate this drug well.

REVIEWERS' COMMENTS:

Reviewer #1 (Remarks to the Author):

revisions are excellent.

Overall the responses clearly answer my questions.

I didn't understand the authors' response to point 3 re Gibb's free energy. This is not a question regarding figure 4; I apologize if my use of the word "calculated" suggested that I was referring to a calculation already present in the paper. It is also not a question of the ion affinities of NKCC1 (although I appreciate the reference for K_o affinity). The question is one of direction, i.e. which way will salt and water be transported under the conditions listed in the original manuscript (the conditions are less clear in the revision lines 83-85). Transport should be inward, i.e cells should swell, if I interpreted the relevant ion concentrations correctly (they are listed in my first set of comments). The authors' response, that the osmotic (water) gradient must overcome the ion gradient, is very interesting and could provide an estimate for the bounds of the stoichiometry of water vs Na/K/Cl transport through NKCC1. I don't think I will be the only reader who will wonder about this, so it would be very useful to squarely address this issue in the manuscript.

Reviewer #2 (Remarks to the Author):

The various points that i raised have been answered satisfactorily. I look forward to the results of future planned experiments.

Norman R Saunders

University of Melbourne

Reviewer #1 (Remarks to the Author):

revisions are excellent.

Overall the responses clearly answer my questions.

I didn't understand the authors' response to point 3 re Gibb's free energy. This is not a question regarding figure 4; I apologize if my use of the word "calculated" suggested that I was referring to a calculation already present in the paper. It is also not a question of the ion affinities of NKCC1 (although I appreciate the reference for K_o affinity). The question is one of direction, i.e. which way will salt and water be transported under the conditions listed in the original manuscript (the conditions are less clear in the revision lines 83-85). Transport should be inward, i.e cells should swell, if I interpreted the relevant ion concentrations correctly (they are listed in my first set of comments). The authors' response, that the osmotic (water) gradient must overcome the ion gradient, is very interesting and could provide an estimate for the bounds of the stoichiometry of water vs Na/K/Cl transport through NKCC1. I don't think I will be the only reader who will wonder about this, so it would be very useful to squarely address this issue in the manuscript.

Answer: We apologize for not having made ourselves clear in our last round of revision. Employing the ion concentrations in the Na⁺-HEPES buffer (+55 mM NaCl or KCl), as the reviewer correctly points out, we predict inward transport of NKCC1 in the choroid plexus in both cases (although four times stronger with the 55 mM KCl). However, it must be noted, we do not know what the ion concentrations are in choroid plexus in bicarbonate-free solutions kept up to 3 hours prior to experiments (the ones used for ion concentration determinations were acutely isolated with the intent purpose of not disturbing the ion gradients).

The speed with which the transporter will run inwards, under these circumstances, is partly dictated by its apparent affinity for K⁺, which is rather low, with a $K_{0.5}$ of around 25 mM in cultured astrocytes (Larsen et al. GLIA, 2014). At 2.5 mM $[K^+]_o$, (as in control solution and in the high NaCl) the inward transport rate is most likely very low (and will thus contribute very little to cotransporter-mediated cell swelling). At the same time, the 100 mOsm osmotic driving force in place to osmotically extract water from the choroid plexus (in a NKCC1-independent manner) will continue to promote cell shrinkage. The total read-out under these circumstances is predicted predominantly to reflect the osmotic 'part' of the water flux through the water permeable, AQP1-containing luminal membrane.

We apologize for not having included a statement to that effect in the manuscript in the last round. We predict that few readers are knowledgeable in this level of detail (the reviewer obviously exempt) and were of the impression that such statement would confuse rather than enlighten. Nevertheless, we have in the revised version included this statement in the discussion: "*While the high $[K^+]_o$ strongly favours inward transport by NKCC1, and thus robust cell swelling, the high $[Na^+]_o$ is, although to a lesser degree, likely to do so as well. However, with the low apparent affinity of NKCC1 for K⁺ (Larsen et al., GLIA, 2014), the inwardly-directed transport under conditions of high $[Na^+]_o$ and low $[K^+]_o$ and the resulting NKCC1-mediated cell swelling will be limited and thus likely masked by the parallel osmotically-induced cell shrinkage.*" We hope we have been able to properly respond to the reviewer's concerns in this manner.